# A quantitative gibberellin signaling biosensor reveals a role for gibberellins in internode specification at the shoot apical meristem

Bihai Shi[1,2,7], Amelia Felipo-Benavent[3,7], Guillaume Cerutti[2], Carlos Galvan-Ampudia[2], Lucas Jilli[3], Geraldine Brunoud[2], Jérome Mutterer[3], Elody Vallet[3], Lali Sakvarelidze-Achard[3], Jean-Michel Davière[3], Alejandro Navarro-Galiano[4], Ankit Walia[5], Shani Lazary[6], Jonathan Legrand[2], Roy Weinstain[6], Alexander M. Jones[5], Salomé Prat[4], Patrick Achard[3] ✉ & Teva Vernoux[2] ✉

Growth at the shoot apical meristem (SAM) is essential for shoot architecture construction. The phytohormones gibberellins (GA) play a pivotal role in coordinating plant growth, but their role in the SAM remains mostly unknown. Here, we developed a ratiometric GA signaling biosensor by engineering one of the DELLA proteins, to suppress its master regulatory function in GA transcriptional responses while preserving its degradation upon GA sensing. We demonstrate that this degradation-based biosensor accurately reports on cellular changes in GA levels and perception during development. We used this biosensor to map GA signaling activity in the SAM. We show that high GA signaling is found primarily in cells located between organ primordia that are the precursors of internodes. By gain- and loss-of-function approaches, we further demonstrate that GAs regulate cell division plane orientation to establish the typical cellular organization of internodes, thus contributing to internode specification in the SAM.

The shoot apical meristem (SAM) at the tip of the shoot axes comprises a stem cell niche whose activity produces lateral organs and stem segments in a modular iterative fashion during the whole plant life. Each of these repetitive units or phytomere includes an internode and lateral organs at a node and an axillary meristem at the leaf axil[1]. The growth and organization of phytomeres change during development. In *Arabidopsis thaliana*, internode growth is inhibited during the vegetative phase and axillary meristems rest dormant at the axils of rosette leaves. Upon floral transition, the SAM turns into an inflorescence meristem, producing elongated internodes and axillary buds that form branches at the axils of cauline leaves, and later flowers without leaves[2]. While substantial progress has been made in our understanding of the mechanisms controlling the initiation of leaves, flowers, and branches, much less is known about how internodes are initiated.

Growth via cell division and expansion is essential for reiterative organogenesis at the SAM. The tetracyclic diterpenoid hormones

[1]College of Agriculture, South China Agricultural University, Guangdong Laboratory for Lingnan Modern Agriculture, 510642 Guangzhou, China. [2]Laboratoire Reproduction et Développement des Plantes, Univ Lyon, ENS de Lyon, CNRS, INRAE, INRIA, 69342 Lyon, France. [3]Institut de biologie moléculaire des plantes, CNRS, Université de Strasbourg, 67084 Strasbourg, France. [4]Centre for Research in Agricultural Genomics, 08193 Cerdanyola, Barcelona, Spain. [5]Sainsbury Laboratory, Cambridge University, Cambridge CB2 1LR, UK. [6]Department of Molecular Biology and Ecology of Plants, Tel Aviv University, Tel Aviv 69978, Israel. [7]These authors contributed equally: Bihai Shi, Amelia Felipo-Benavent. ✉e-mail: patrick.achard@ibmp-cnrs.unistra.fr; teva.vernoux@ens-lyon.fr

gibberellins (GA) are key growth regulators[3–5], with a crucial role in many embryonic and post-embryonic developmental processes[6]. Central to the GA signaling pathway are the five DELLA proteins, GIB-BERELLIC ACID INSENSITIVE (GAI), REPRESSOR OF GA1-3 (RGA), and RGA-Like (RGL) 1-3[7]. These nuclear proteins are composed of an N-terminal DELLA/TVHYNP domain and of a GRAS domain. The GRAS domain allows DELLAs to interact with diverse transcription factors and transcriptional regulators and to suppress growth by modulating their activity[7]. The binding of GA to the GIBBERELLIN INSENSITIVE DWARF1 (GID1) GA receptor promotes GID1 interaction with the N-terminal domain of DELLAs, triggering DELLA degradation by the ubiquitin-dependent proteasome pathway[7–10], and in turn, de-repressing GA responses. Despite the relatively specialized role of the GRAS and DELLA/TVHYNP domains, residues required for DELLA degradation and partner protein interaction are widely distributed within the protein sequence.

The identification of genes encoding GA catabolic enzymes as direct targets of the class I KNOX meristem identity regulators has led to the proposal that GA levels are low in the SAM cells, while high GA concentrations trigger the growth of lateral organs[3–5]. Low GA levels could then contribute to SAM maintenance[3]. However, more recent analysis indicates also that GAs promote the increase in SAM size during the floral transition by regulating the division and expansion of inner SAM cells[11,12], in a similar manner as in the root[13,14]. DELLAs were likewise found to limit meristem size by directly regulating the expression of the cell-cycle inhibitor KRP2 in the internal part of the SAM (the rib zone)[12]. Together, these findings support a role for GA in positively regulating cell division in the inner tissues of the SAM, and thus SAM size. At the same time, several genes encoding GA biosynthetic and catabolic enzymes are expressed specifically in lateral organs[4,11], illustrating a complex spatiotemporal GA distribution in the SAM to likely fulfill different functions.

Accessing spatiotemporal GA distribution has been instrumental in better understanding the functions of these hormones in different tissues and at various developmental stages. Visualization of degradation of an RGA-GFP fusion expressed under its own promoter provided key information on the regulation of GA global levels in the root[15,16]. However, *RGA* is expressed differentially in tissues[17] and this expression is regulated by GA[18]. Differential expression from the *RGA* promoter thus potentially contributes to the fluorescence pattern observed with RGA-GFP, making this approach not quantitative. Accumulation of GA in the root endodermis and regulation of their cellular level via GA transport was later discovered by using bioactive fluorescein (Fl)-tagged GA[19,20]. More recently, the nlsGPS1 GA FRET sensor revealed that GA levels are correlated with cell elongation in roots, stamen filaments, and dark-grown hypocotyls[21]. However, as we have seen, GA concentration is not the only parameter controlling GA signaling activity as it depends on a complex perception process. Here, building on the knowledge of DELLAs and the GA signaling pathway, we report on the engineering and characterization of a degradation-based GA signaling ratiometric biosensor. To design this quantitative biosensor, we used a mutated yet GA-sensitive RGA fused to a fluorescent protein and expressed ubiquitously in tissues, together with a GA-insensitive fluorescent protein. We show that the mutated RGA protein fusion does not interfere with endogenous GA signaling when expressed ubiquitously and that the biosensor allows quantifying the signaling activity resulting from the contribution of GA and of the perception machinery processing the GA signal with a high spatiotemporal resolution. We used this biosensor to map the spatiotemporal distribution of GA signaling activity and to quantitatively analyze how GAs regulates cell behavior in the SAM epidermis. We demonstrate that GAs regulate the orientation of division planes of SAM cells located between organ primordia, therefore specifying the typical cellular organization of internodes.

## Results

### Modifying the RGA protein for sensor construction

DELLA degradation results from the processing of the information from GA by a complex perception process involving GID1 receptors and ubiquitination and is then a readout of the GA signaling activity. We thus aimed at generating a degradation-based GA signaling biosensor by engineering a DELLA protein to meet two criteria, (i) a specific degradation upon GA perception; and (ii) a minimal interference with GA signaling. To do so, we modified DELLAs to preserve the interaction with GID1 while abolishing interactions with partner transcription factors, by introducing mutations in the GRAS domain. Among the 5 *Arabidopsis* DELLAs, RGA displays one of the highest GA-dependent degradation rate[22], and RGA-GFP fusions are widely used as GA signaling reporter[8]. Leveraging on the results of GRAS domain mutant analyses in rice[23] and *Arabidopsis*[24,25], we generated four modified RGA versions (RGA^m1: G218A, V219A, R220A; RGA^m2: H471A, Y472A, Y473A; RGA^m3: S578Stop; and RGA^m4: S578D) and tested their ability to meet the above-defined criteria (Fig. 1a and Supplementary Fig. 1a). We first used a yeast-two-hybrid (Y2H) assay to test the binding capacity of these modified candidates with three well known DELLA-interacting partners, JAZ1, TCP14 and IDD2[26]. While RGA^m1 had a minor effect on interactions, RGA^m2, RGA^m3, and RGA^m4 lost their capacity to interact with these partners (Fig. 1b and Supplementary Fig. 1b). In addition, RGA^m2, RGA^m3 and RGA^m4 were able to bind GID1 in the presence of GA, thus suggesting that these DELLA candidates are still degraded in response to GA (Fig. 1c and Supplementary Fig. 1c). To assess this possibility, we explored GA-dependent degradation of RGA^m2, RGA^m3 and RGA^m4 fused to GFP in transient expression assays. These three candidates were degraded after GA treatment (Fig. 1d and Supplementary Fig. 1d), RGA^m2-GFP having the fastest degradation kinetics although it was slightly more stable than RGA-GFP. Noteworthy, RGA^m2 harbors three amino acid substitutions in the GRAS PFYRE motif (Fig. 1a), which is highly conserved in all plant DELLAs[26]. Therefore, this variant is likely to have similar properties in plant species other than *Arabidopsis*.

Based on the above results, we selected RGA^m2 for further analysis. We next showed that RGA^m2 is unable to bind with the BZR1 DELLA-interacting partner in yeast and a larger screen confirmed that these mutations abolish interactions with practically all known DELLA partners (Fig. 1b and Supplementary Table 1). Moreover, co-immunoprecipitation studies demonstrated that binding of RGA^m2 to IDD2 or TCP14 is also strongly reduced *in planta* compared to RGA (Fig. 1e). Finally, transient expression assays confirmed that, while RGA represses BZR1 and TCP14 transcriptional activities, RGA^m2 did not (Fig. 1f, g). Taken together, these data indicate that the expression of RGA^m2 in plants, hereafter named mRGA for simplicity might have a limited impact on GA signaling. Thus, mRGA constitutes a suitable DELLA variant candidate for engineering a degradation-based biosensor that monitors GA signaling activity and more specifically the combinatorial effect of GA and of its complex perception machinery.

### Engineering a GA signaling sensor

To create a ratiometric GA signaling sensor, we fused the mRGA protein to the fast-maturing yellow fluorescent protein VENUS[27] and co-expressed this fusion protein together with a nuclear-localized non-degradable reference protein, TagBFP-NLS, under a 2.5 kb *pUBQ10*[28] or *pRPS5a*[29,30] constitutive promoter. We used the 2A self-cleaving peptide to allow for a stoichiometric production of both fluorescent proteins, enabling quantification of GA signaling activity using fluorescence intensity ratio between mRGA-VENUS and TagBFP[31,32] (Fig. 2a). We named the sensor lines qmRGA (quantitative mRGA) and first analyzed their TagBFP fluorescence pattern in vegetative and reproductive tissues (Fig. 2b and Supplementary Fig. 2a, b). TagBFP fluorescence of *pUBQ10::qmRGA* lines was homogeneously distributed

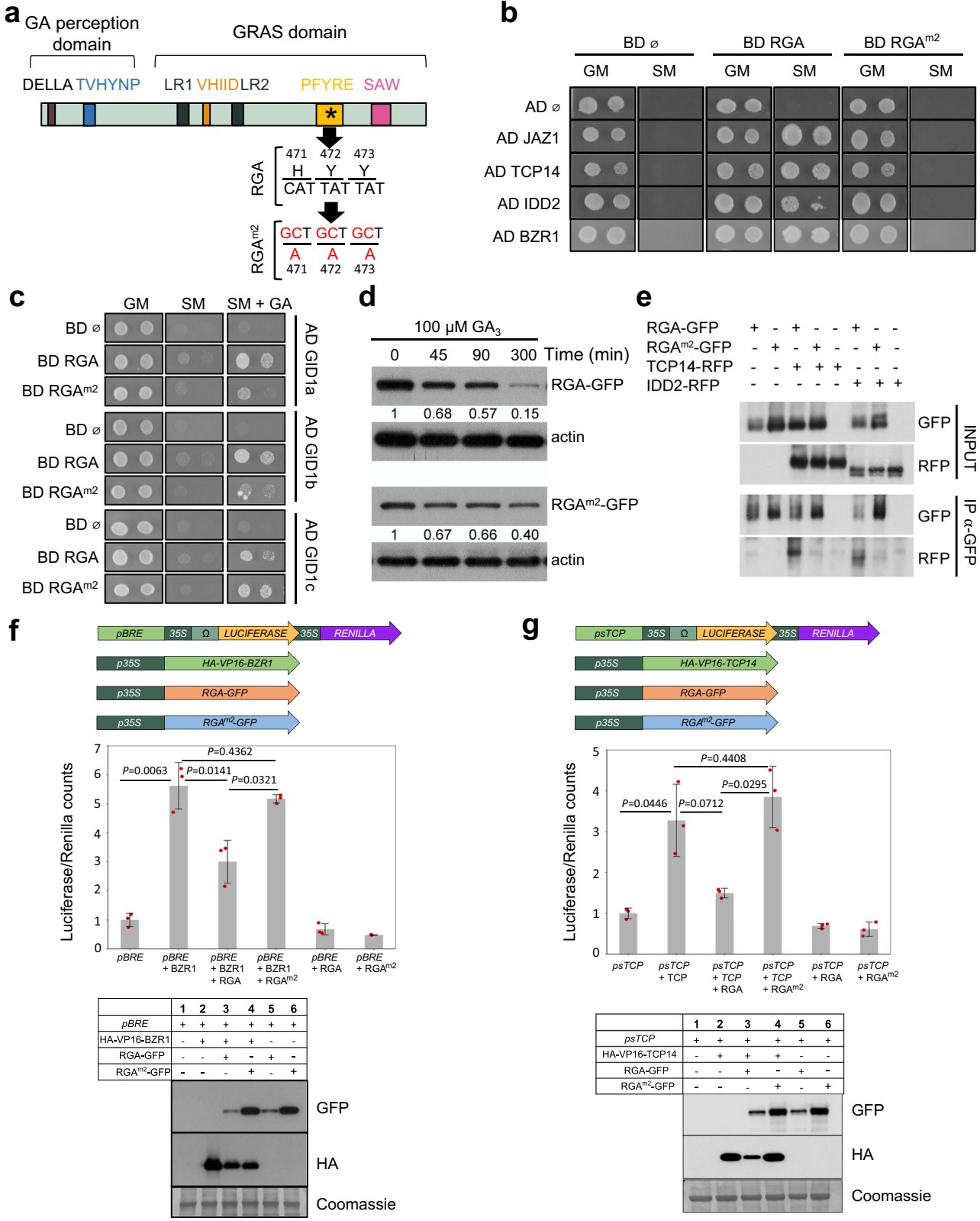

in hypocotyl, the vegetative shoot meristem, cotyledons, and roots (except in the root tip) but the TagBFP signal was unevenly distributed in the inflorescence SAM, with a stronger signal in organ boundaries (Supplementary Fig. 2a). Conversely, the TagBFP signal was strong in the root tip and the vegetative SAM in *pRPS5a::qmRGA* lines (Fig. 2b), and showed a homogenous distribution in the inflorescence SAM (Supplementary Fig. 2b). Hence, the choice of one of these constructs

depends on the analyzed tissue: we used *pUBQ10::qmRGA* in subsequent experiments in seedlings except root tip, and *pRPS5a::qmRGA* for experiments performed in root tip and inflorescence SAM.

To test that qmRGA activity is indeed not interfering with signaling activity and thus with plant growth, we first investigated the effects of exogenous GA and paclobutrazol (Pac; an inhibitor of GA biosynthesis) treatments on the growth of qmRGA plants. We found that

**Fig. 1 | The modified RGA^m2 protein is an inactive DELLA that remains sensitive to GA. a** Schematic representation of the domain structure of a typical DELLA protein. Conserved histidine (H471), tyrosine (Y472), and tyrosine (Y473) residues were mutated into alanine (A) to obtain the modified RGA^m2 protein. The nucleic acids and amino acids mutated in RGA^m2 are indicated in red. **b** Yeast two-hybrid (Y2H) assays in which RGA and RGA^m2 were tested pairwise with four known DELLA-interacting partners: JAZ1, TCP14, IDD2 and BZR1. Empty pGBKT7 and pGADT7 vectors were included as negative controls. Photos show the growth of the yeast on control media (GM) and on selective media (SM). **c** Pairwise Y2H interaction assays between RGA or RGA^m2 and the three GA receptors GID1a, GID1b and GID1c. Photos show the growth of the yeast on control media (GM), selective media (SM), and SM media supplemented with 100 µM GA₃. **d** Time-course analysis of GA-induced degradation of RGA (upper panel) and RGA^m2 protein (lower panel). Immunodetection of RGA-GFP and RGA^m2-GFP protein in *35S::RGA-GFP* and *35S::RGA^m2-GFP N. benthamiana* agro-infiltrated leaves treated with 100 mM cyclo-heximide (CHX) and 100 µM GA₃ for the indicated times. Numbers indicate RGA-GFP and RGA^m2-GFP levels relative to actin levels, used as loading control. The experiment was repeated twice with similar results. **e** Co-immunoprecipitation assays between RGA or RGA^m2 and IDD2 or TCP14. Protein extracts from different combinations of *N. benthamiana* agro-infiltrated leaves with *35S::RGA-GFP*, *35S::RGA^m2-GFP*, *35S::IDD2-RFP*, and *35S::TCP14-RFP* were immunoprecipitated with anti-GFP antibodies. The co-immunoprecipitated protein (IDD2-RFP and TCP14-RFP) was detected by anti-RFP antibodies. The experiment was repeated twice with similar results. **f, g** Effect of RGA and RGA^m2 on BZR1 (**f**) and TCP14 (**g**) transcriptional activities in *N. benthamiana* agro-infiltrated leaves with a combination of BZR1, TCP14, RGA, and RGA^m2 effector constructs and corresponding Luciferase/Renilla reporter constructs, as indicated (top panels). BZR1 and TCP14 have been fused to VP16 transcriptional activator domain in this experiment. Transcriptional activities are represented as the ratio of Luciferase and Renilla (used as internal control) activities, relative to the value obtained for the reporter construct alone that was set to 1. Data are means ± SD of three biological replicates. *P*-values were calculated in R using a two-tailed Welch *t*-test. Bottom panels: immunodetection of RGA-GFP, RGA^m2-GFP, HA-VP16-BZR1, and HA-VP16-TCP14 from *N. benthamiana* agro-infiltrated leaves used for transcriptional activity assays. These experiments were repeated three times with similar results.

the hypocotyl length of qmRGA seedlings was similar to that of wild-type under mock conditions and upon GA or Pac treatment, although hypocotyls were slightly longer after GA treatment (Fig. 2c). Similarly, shoot development and plant fertility were not significantly affected in qmRGA plants (Fig. 2d and Supplementary Fig. 2c). Last, we showed that when mRGA in the qmRGA construct is replaced by d17RGA, a mutant version of RGA that is fully insensitive to GA[33], the resulting qd17RGA plants exhibited a severe dwarf phenotype reminiscent of GA-insensitive mutants. By contrast, qd17mRGA plants, expressing a mutated version of mRGA to which the d17 mutation was added, had similar rosette size and height to wild-type plants (Supplementary Fig. 2d–i). Altogether, these results demonstrate that qmRGA negligibly interferes with plant growth and GA responses.

Next, we assessed whether qmRGA can detect changes in signaling activity resulting from changes in GA levels. Consistent with the above transient expression assays, GA treatment induced the degradation of mRGA in *pRPS5a::qmRGA* seedlings, although the protein tends to be slightly more stable than RGA (Fig. 2e, f). Accordingly, while the TagBFP signal was unaffected in hypocotyls of *pUBQ10::qmRGA* seedlings upon GA application, fluorescence of the mRGA-VENUS sensor element was substantially reduced after this treatment (Fig. 2g). Similar results were observed in qmRGA root tips for which GA and Pac application respectively reduced and increased mRGA-VENUS signal (Supplementary Fig. 3a, b). In contrast, VENUS fluorescence was much less affected by the treatments in the RGA^m3-VENUS and RGA^m4-VENUS lines, consistently with the lower GA-dependent degradation rate of these variants (Supplementary Fig. 3c–f).

Since mRGA-VENUS is degraded upon GA sensing in qmRGA plants, mRGA-VENUS fluorescence is negatively related to GA signaling, while−mRGA-VENUS/TagBFP is positively related to GA signaling and accounts for possible variations in promoter activity. In further image analyses, we then used 3− (mRGA-Venus/TagBFP) to have a positive proxy for GA signaling activity that fully covers the range of values in VENUS/TagBFP fluorescence ratio that we measure (hereafter named "GA signaling"; Fig. 2a; see also Supplementary Methods). This quantitative approach confirmed statistically significant changes in GA signaling in hypocotyls, with an increase and decrease respectively upon GA and Pac treatments compared to untreated seedlings (Fig. 2g, h and Supplementary Fig. 4a–d). Exogenous GA and Pac treatments induced similar responses in the SAM, although the effect of Pac was less pronounced than in hypocotyl (Supplementary Fig. 5). Furthermore, GA signaling activity increased in the SAM with both exogenous GA concentration and treatment duration (Supplementary Fig. 4e–i), showing that qmRGA is suitable to be used as a GA signaling sensor in the SAM as in outer cell layers of hypocotyls and root tips (Supplementary Fig. 3a, b).

Finally, we asked whether qmRGA is able to report changes in endogenous GA levels using growing hypocotyls. We previously showed that nitrate promotes growth by increasing GA synthesis and in turn DELLA degradation[34]. Accordingly, we observed that hypocotyl length of *pUBQ10::qmRGA* seedlings grown on adequate nitrate supply (10 mM NO₃⁻) was significantly longer compared to those grown on nitrate-deficient conditions (Supplementary Fig. 6a). Consistent with the growth response, GA signaling was higher in hypocotyls of seedlings grown with 10 mM NO₃⁻ compared with those grown in absence of nitrate (Supplementary Fig. 6b, c). Thus, qmRGA also allows monitoring changes in GA signaling resulting from endogenous changes in GA concentration.

## qmRGA fluorescence depends on GA receptor activity

To ask if the GA signaling activity reported by qmRGA depends on both GA concentration and GA perception as expected from the sensor design, we analyzed the expression of the three *GID1* receptors in vegetative and reproductive tissues. In seedlings, GID1-GUS reporter lines showed that *GID1a* and *c* are highly expressed in cotyledons (Fig. 3a–c). Moreover, all three receptors are expressed in leaves, lateral root primordia, root tip (excluding root cap for *GID1b*), and vasculature (Fig. 3a–c). In inflorescence SAM, we only detected a GUS signal for GID1b and 1c (Supplementary Fig. 7a–c). In situ hybridization confirmed these expression patterns and further demonstrated that *GID1c* is homogenously expressed at a low level in the SAM while *GID1b* shows higher expression at the SAM periphery (Supplementary Fig. 7d–l). A *pGID1b::2xmTQ2-GID1b* translational fusion further revealed a graded *GID1b* expression ranging from low or no expression in the SAM center to high expression in organ boundaries (Supplementary Fig. 7m). Thus, GID1 receptors are unevenly distributed across and within tissues. In a subsequent experiment, we also observed that overexpressing *GID1* (*pUBQ10::GID1a-mCherry*) enhanced qmRGA sensitivity to external GA application in hypocotyls (Fig. 3d, e). By contrast, the fluorescence measured from qd17mRGA in hypocotyls was insensitive to GA₃ treatment (Fig. 3f, g). For these two assays, seedlings were treated with a high concentration of GA (100 µM GA₃) in order to assess the fast behavior of the sensor when the capacity to bind the GID1 receptors is enhanced or lost. Taken together, these results confirm that the qmRGA biosensor reports for the combinatorial action of GA and GA perception and suggest that differential expression of GID1 receptors notably can modulate the emission ratio of the sensor.

## A GA signaling map in the shoot apical meristem

The distribution of GA signaling within the SAM has remained elusive so far. Thus, we used plants expressing qmRGA together with the

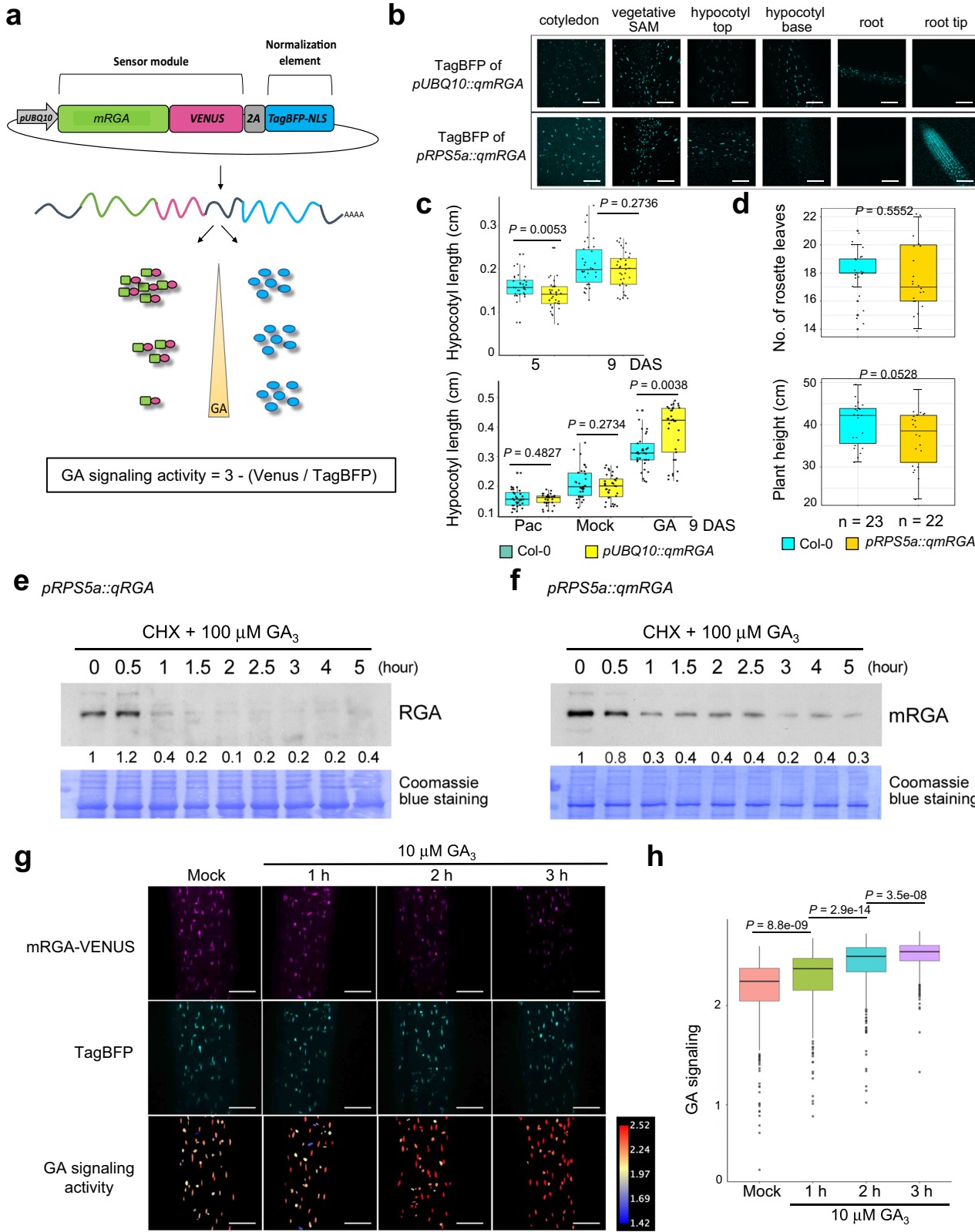

*pCLV3::mCherry-NLS*[35] stem cell reporter to compute a high-resolution quantitative map of GA signaling activity, focusing on the L1 layer (epidermis; Fig. 4a, b, see Methods and Supplementary Methods) given the key role of L1 in controlling growth at the SAM[36]. Here, the expression of *pCLV3::mCherry-NLS* provided a fixed geometric reference for analyzing the spatiotemporal distribution of GA signaling activity[37]. Although GAs were proposed to be required for lateral organ

development[4], we observed that GA signaling was lower in flower primordia (P) from the $P_3$ stage onward (Fig. 4a, b), while young $P_1$ and $P_2$ primordia had intermediate activity similar to the one found in the central zone (Fig. 4a, b). Higher GA signaling activity was found in the boundaries of organ primordia, starting from $P_1/P_2$ (on the lateral sides of the boundary) and culminating from $P_4$, and in all peripheral zone cells located between primordia (Fig. 4a, b and Supplementary

**Fig. 2 | The qmRGA sensor monitors changes in GA levels. a** Schematic representation of the qmRGA construct composed of two elements: the sensor module (mRGA-VENUS), and the normalization element (TagBFP fused with a nuclear localization signal (NLS)). The two elements are linked by a 2A self-cleaving peptide, and driven by the same promoter allowing stoichiometric expression. GA signaling activity is measured as 3 minus the ratio between VENUS and TagBFP signal intensities. **b** TagBFP expression pattern of *pUBQ10::qmRGA* and *pRPS5a::qmRGA* sensors monitored in cotyledon, vegetative SAM, hypocotyl, and root of 7-day-old seedlings. The experiment was repeated twice with similar results. **c** Boxplot representations of the hypocotyl length of wild-type (Col-0) and *pUBQ10::qmRGA* seedlings. Upper panel: hypocotyl length at 5 (Col-0, $n = 33$ biological independent seedlings; qmRGA, $n = 36$ biological independent seedlings) and 9 days (Col-0, $n = 34$ biological independent seedlings; qmRGA, $n = 35$ biological independent seedlings) after sowing (DAS). Lower panel: hypocotyl length of 9-day-old seedlings grown for 4 days on MS media and then transferred to MS media (Mock; Col-0, $n = 34$ biological independent seedlings; qmRGA, $n = 35$ biological independent seedlings), supplemented with 5 µM Pac (Col-0, $n = 33$ biological independent seedlings; qmRGA, $n = 30$ biological independent seedlings) or 10 µM GA$_3$ (Col-0, $n = 33$ biological independent seedlings; qmRGA, $n = 33$ biological independent seedlings). Center lines show the medians and box limits indicate the 25th and 75th percentiles. Whiskers indicate minima and maxima as determined using the R software. *P*-values are determined with the R software using a two-tailed Welch *t*-test. **d** Boxplot representations of the number of rosette leaves (upper panel) and

final plant height (lower panel) of wild-type (Col-0, $n = 23$) and *pRPS5a::qmRGA* adult plants ($n = 22$). Center lines show the medians and box limits indicate the 25th and 75th percentiles. Whiskers indicate minima and maxima as determined using the R software. *P*-values are from two-sided Student's *t*-tests. **e, f** Time-course analysis of GA-induced RGA and mRGA degradation. 7-day-old *pRPS5a::qRGA* (**e**) and *pRPS5a::qmRGA* seedlings (**f**) were treated with 100 µM cycloheximide (CHX) and 100 µM GA$_3$. At indicated time points, total proteins were extracted and analyzed by immunoblot using RGA antibodies. Numbers indicate the fold increase in RGA and mRGA protein levels relative to the blue-stained protein signal. The experiment was repeated twice with similar results. **g** Representative confocal images of mRGA-VENUS and TagBFP signals and corresponding heatmap representation of GA signaling activity in hypocotyls of 5-day-old *pUBQ10::qmRGA* seedlings treated with 10 µM GA$_3$ for the time indicated (and mock control). The experiment was repeated twice with similar results. **h** Boxplot representation of GA signaling activity (Mock, $n = 467$ nuclei examined over ten independent seedlings; 1 h, $n = 331$ nuclei examined over eight independent seedlings; 2 h, $n = 486$ nuclei examined over nine independent seedlings; 2 h, $n = 550$ nuclei examined over nine independent seedlings) measured in *pUBQ10::qmRGA* seedling hypocotyls grown in the same conditions as in (**g**), indicated by different colors. Center lines show the medians and box limits indicate the 25th and 75th percentiles. Whiskers indicate minima and maxima as determined using the R software. *P*-values are from Kruskal–Wallis tests. The experiment was repeated twice with similar results. Scale bars = 100 µm.

Fig. 8a, b). This higher GA signaling activity was not only monitored in the epidermis but also in the L2 and upper L3 layers (Supplementary Fig. 8b). The GA signaling pattern detected with qmRGA in the SAM was also consistent over time (Supplementary Fig. 8c–f, k). While the *qd17mRGA* construct was systematically silenced in the SAM of T3 plants in 5 independent lines we characterized in depth, we could analyze the fluorescence pattern obtained with *pRPS5a::VENUS-2A-TagBFP* construct (Supplementary Fig. 8g–j, l). Only small variations in fluorescence ratio were detected in the SAM from this control line, except at the center of the SAM where we robustly observed an unexpected decrease in VENUS related to TagBFP. This confirms that the signaling pattern observed with qmRGA reflects GA-dependent degradation of mRGA-VENUS but also that qmRGA might overestimate GA signaling activity at the center of the meristem. Taken together, our results reveal a GA signaling pattern mostly mirroring primordia distribution. This inter-primordia-region (IPR) distribution results from the progressive establishment of a high GA signaling activity between developing primordia and the central zone, while in parallel GA signaling activity decreases in primordia (Fig. 4c, d).

The distribution of GID1b and GID1c receptors (see above) suggests that differential expression of GA receptors contributes to shaping the GA signaling activity pattern in the SAM. We wondered if the differential accumulation of GA could also be involved. To investigate this possibility, we used the nlsGPS1 GA FRET sensor[21]. An increased emission ratio was detected in nlsGPS1 SAMs treated with 10 µM GA$_{4+7}$ for 100 min (Supplementary Fig. 9a–e), indicating that nlsGPS1 responds to changes in GA concentration in the SAM as it does in roots[21]. The spatial distribution of the nlsGPS1 emission ratio indicates that GA levels are relatively low in the SAM external layers, but it shows that they are elevated in the center of the SAM and in the boundaries (Fig. 4e and Supplementary Fig. 9a, c). This suggests that GAs are also distributed in the SAM with a spatial pattern comparable to the one revealed by qmRGA. As a complementary approach, we also treated SAMs with fluorescent GAs (GA$_3$-, GA$_4$-, GA$_7$-Fl) or with Fl alone as a negative control. The Fl signal was distributed in the whole SAM, including the central zone and primordia, although with lower intensity (Fig. 4j and Supplementary Fig. 10d). In contrast, all the three GA-Fls specifically accumulated in primordia boundaries and, to different degrees, in part of the rest of the IPR, with GA$_7$-Fl accumulating in the largest domain in the IPR (Fig. 4k and Supplementary Fig. 10a, b). Fluorescence intensity quantification demonstrated a higher IPR to non-IPR intensity ratio in the GA-Fl-treated SAMs, compared to

Fl-treated SAMs (Fig. 4l and Supplementary Fig. 10c). Taken together, these results suggest that GAs are present at higher levels in the IPR cells closest to organ boundaries. This indicates that the SAM GA signaling activity pattern results from both differential expression of the GA receptors and from the differential accumulation of GA in IPR cells closest to organ boundaries. Our analysis thus identifies an unexpected spatiotemporal GA signaling pattern, with lower activity in the center of the SAM and in primordia, while activity is elevated in the IPR of the peripheral zone.

## Correlation analyses suggest a role for GA signaling in cell division plane orientation in the shoot apical meristem

To understand the role of differential GA signaling activity at the SAM, we analyzed the correlation between GA signaling activity, cell expansion, and cell division using time-lapse live imaging of *qmRGA pCLV3::mCherry-NLS* SAMs. Given the role of GA in growth regulation, a positive correlation was expected with cell expansion parameters. We thus first compared maps of GA signaling activity to those of cell surface growth rate (as a proxy for cell expansion intensity for a given cell and daughter cells if it divides) and growth anisotropy, which measures the directionality of cell expansion (here also for a given cell and daughter cells if it divides; Fig. 5a, b, see Methods and Supplementary Methods). Our cell surface growth intensity map of the SAM was consistent with previous observations[38,39], with a minimal growth rate in boundaries and a maximal rate in developing flowers (Fig. 5a). A principal component analysis (PCA) showed an anti-correlation between GA signaling activity and cell surface growth intensity (Fig. 5c). It further showed that the main axis of variability, encompassing GA signaling input and growth intensity, was orthogonal to the direction defined by high expression of *CLV3*, which argues in favor of excluding cells from the center of the SAM in the rest of the analysis. Spearman correlation analyses confirmed the PCA results (Fig. 5d), suggesting that higher GA signaling in the IPR does not lead to higher cell expansion. However, the correlation analysis demonstrated a mild positive association between GA signaling activity and growth anisotropy (Fig. 5c, d), suggesting that higher GA signaling in the IPR acts on cell growth orientation and possibly cell division plane positioning.

Thus, we next studied the correlation between GA signaling and cell division activity, by identifying newly formed cell walls during a time-course analysis (Fig. 5e). This methodology allows us to measure both the frequency and orientation of cell divisions. Strikingly, we

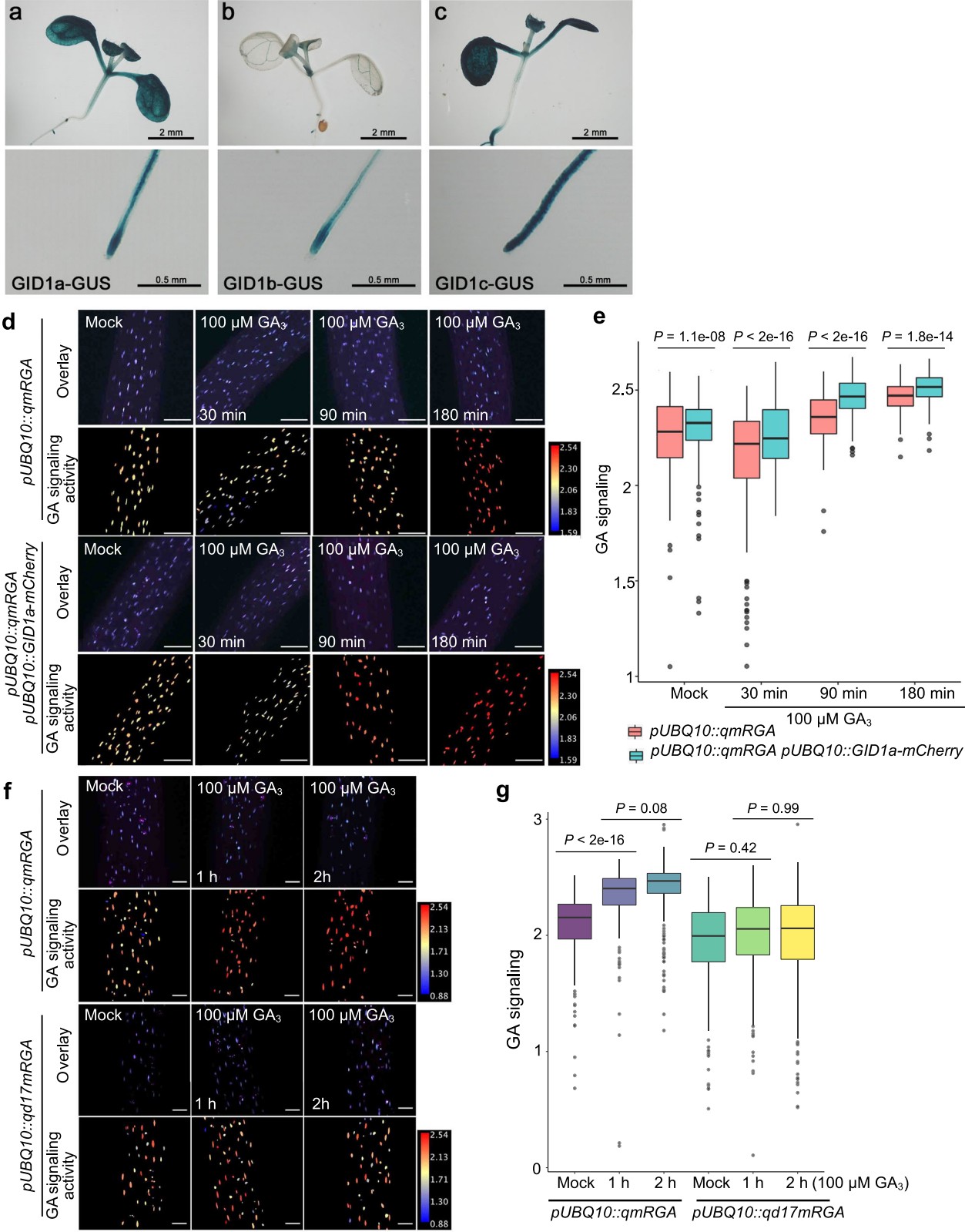

found that cell division frequency was similar in the IPR and the rest of the SAM (non-IPR, Fig. 5f), showing that differences in GA signaling between IPR and non-IPR cells do not have a major effect on cell division. This, together with the positive correlation between GA signaling and growth anisotropy, led us to ask whether GA signaling activity could act on cell division plane orientation. We measured the orientation of new cell walls as the acute angle relative to the radial axis

connecting the center of the meristem to the center of new cell walls (Fig. 5e–i), and observed that cells had a clear tendency to divide at angles closer to 90° relative to the radial axis, with the highest frequency observed at 70–80° (23.28%) and 80–90° (22.62%) (Fig. 5e, i), i.e., corresponding to cell divisions oriented in the circumferential/transverse direction (Fig. 5h). To explore the contribution of GA signaling to this cell division behavior, we analyzed separately the cell

**Fig. 3 | The qmRGA sensor signal depends on GA receptor activity.**
**a–c** Expression pattern of *GID1a*, *GID1b* and *GID1c* in shoots (upper panel) and root tips (lower panel) of 7-day-old *pGID1a::GID1a-GUS* (**a**), *pGID1b::GID1b-GUS* (**b**) and *pGID1c::GID1c-GUS* (**c**) seedlings. **d, f** Overlay of VENUS and TagBFP maximum intensity projection (upper row), and corresponding heatmap representation of GA signaling activity (lower row) in hypocotyls of 5-day-old *pUBQ10::qmRGA* and *pUBQ10::qmRGA pUBQ10::GID1a-mCherry* seedlings treated with 100 µM GA$_3$ for 30, 9,0 and 180 min (**d**), and of *pUBQ10::qmRGA* and *pUBQ10::qd17mRGA* seedlings treated with 100 µM GA$_3$ for 1 and 2 h (**f**) and mock controls. **e** Boxplot representation of GA signaling activity (Mock, *n* = 65 nuclei examined over three independent seedlings; 100 µM GA$_3$ for 30 min, *n* = 163 nuclei examined over four independent seedlings; 90 min, *n* = 129 nuclei examined over three independent seedlings; 180 min, *n* = 142 nuclei examined over three independent seedlings) measured in *pUBQ10::qmRGA* seedling hypocotyls, and (Mock, *n* = 150 nuclei examined over four independent seedlings; 100 µM GA$_3$ for 30 min, *n* = 182 nuclei examined over four independent seedlings; 90 min, *n* = 251 nuclei examined over five independent seedlings; 180 min, *n* = 101 nuclei examined over three independent seedlings) measured in *pUBQ10::qmRGA pUBQ10::qmRGA pUBQ10::GID1a-mCherry* seedling hypocotyls grown in the same conditions as in (**d**), indicated by different colors. Center lines show the medians and box limits indicate the 25th and 75th percentiles. Whiskers indicate minima and maxima as determined using the R software. *P*-values are from Kruskal−Wallis tests. The experiment was repeated three times with similar results. **g** Boxplot representation of GA signaling activity (Mock, *n* = 332 nuclei examined over eight independent seedlings; 100 µM GA$_3$ for 1 h, *n* = 329 nuclei examined over seven independent seedlings; 2 h, *n* = 343 nuclei examined over seven independent seedlings) measured in *pUBQ10::qmRGA* seedling hypocotyls, and (Mock, *n* = 293 nuclei examined over nine independent seedlings; 100 µM GA$_3$ for 1 h, *n* = 421 nuclei examined over ten independent seedlings; 2 h, *n* = 343 nuclei examined over ten independent seedlings) measured in *pUBQ10::qd17mRGA* seedling hypocotyls grown in the same conditions as in (**f**), indicated by different colors. Center lines show the medians and box limits indicate the 25th and 75th percentiles. Whiskers indicate minima and maxima as determined using the R software. *P*-values are from Kruskal−Wallis tests. The experiment was repeated twice with similar results. Scale bars = 100 µm.

division parameters in the IPR and non-IPR (Fig. 5i). We observed that the distribution of cell division angles was different for IPR compared to non-IPR cells or the cells from the entire SAM, with IPR cells showing a higher proportion of transverse/circumferential cell divisions, i.e. 70–80° and 80–90° (the corresponding proportion is 33.86% and 30.71%, respectively) (Fig. 5i). Thus, our observations reveal a link between high GA signaling and cell division plane orientation close to the circumferential direction that parallels the correlation between GA signaling activity and growth anisotropy (Fig. 5c, d). To further establish spatial conservation of this link, we measured the orientation of division planes in IPR cells around primordia starting at stage P$_3$, given that the highest GA signaling activity is detected in this region from stage P$_4$ (Fig. 4). The division angles of the IPR around P$_3$ and P$_4$ did not show statistically significant differences, although an increase in the frequency of transverse cell divisions was observed in the IPR around P$_4$ (Fig. 5j). Differences in cell division plane orientation were however statistically significant in IPR cells around P$_5$, in which the frequency of transverse cell divisions was drastically increased (Fig. 5j). Taken together, these results suggest that GA signaling could control orientation of cell division in the SAM coherently with previous reports[40,41], with high GA signaling likely inducing a transverse orientation of cell divisions in the IPR.

## GA signaling activity positively regulates transverse cell divisions in the shoot apical meristem
Cells in the IPR are expected not to be incorporated into primordia but rather in the internodes[2,42,43]. A transverse orientation of cell divisions in the IPR could generate the typical organization in parallel longitudinal cell files of the internode epidermis. Our observations above indicate that GA signaling is likely to act in this process by regulating cell division orientation.

Loss-of-function of multiple *DELLA* genes leads to constitutive GA response, and thus *della* mutants could be used to test this hypothesis[44]. We first analyzed the expression patterns of the five *DELLA* genes in the SAM. Transcriptional fusion GUS lines[45] showed that *GAI, RGA, RGL1*, and, to a much lesser extent, *RGL2* are expressed in the SAM (Supplementary Fig. 11a–d). In situ hybridization further showed that *GAI* mRNA specifically accumulates in primordia and developing flowers (Supplementary Fig. 11e). *RGL1* and *RGL3* mRNA were detected throughout the SAM dome and in older flowers, while *RGL2* mRNA was more abundant in the boundary regions (Supplementary Fig. 11f–h). Confocal imaging of *pRGL3::RGL3-GFP* SAM confirmed the expression observed with in situ hybridization and showed that the RGL3 protein accumulates in the central part of the SAM (Supplementary Fig. 11i). Using a *pRGA::GFP-RGA* line, we also found that the RGA protein accumulates in the SAM, but its abundance is reduced in boundaries starting from P$_4$ (Supplementary

Fig. 11j). Notably, the expression patterns of *RGL3* and *RGA* are compatible with a higher GA signaling activity in the IPR, as detected with qmRGA (Fig. 4). In addition, these data indicate that all *DELLAs* are expressed in the SAM and that collectively, their expressions cover the entire SAM.

We next analyzed the cell division parameters in wild-type (L*er*, control) and quintuple (global) *gai-t6 rga-t2 rgl1-1 rgl2-1 rgl3-4 della* mutant SAMs (Fig. 6a, b). Interestingly, we observed a statistically significant change in frequency distribution of cell division angles in the SAM of global *della* mutant compared to wild-type (Fig. 6c). This change in the global *della* mutant resulted from an increase in frequency of 80–90° angles (34.71 % vs 24.55 %) and to a lesser extent of 70–80° angles (23.78 % vs 20.18 %), i.e. corresponding to transverse cell divisions (Fig. 6c). The frequency of non-transverse divisions (0–60°) was also lower in global *della* mutant (Fig. 6c). The increased occurrence of transverse cell divisions was easily visible in the SAM of global *della* mutant (Fig. 6b). The frequency of transverse cell divisions in the IPR was also higher in global *della* mutant compared to wild-type (Fig. 6d). Outside of the IPR region, distribution of the angle of cell division was more homogeneous in wild-type, while in the global *della* mutant it was skewed toward tangential divisions, as in the IPR (Fig. 6e). We also quantified cell division orientation in the SAM of quintuple *ga2-oxidase* (*ga2ox*) mutants (*ga2ox1-1, ga2ox2-1, ga2ox3-1, ga2ox4-1*, and *ga2ox6-2*), a GA inactivation mutant background in which GA accumulates. Consistent with an increase in GA levels, the quintuple *ga2ox* mutant has a bigger inflorescence SAM than Col-0 (Supplementary Fig. 12a, b) and, compared to Col-0, the quintuple *ga2ox* SAMs showed a significantly different distribution of cell division angles with increases of the frequencies of angles from 50 to 90°, i.e., again skewed toward tangential divisions (Supplementary Fig. 12a–c). We thus show that both constitutive activation of GA signaling and accumulation of GA induce transverse cell division both in the IPR and the rest of the SAM.

We then tested the effect of inhibiting GA signaling specifically in the IPR. To do so, we used the *CUP-SHAPED COTYLEDON 2* (*CUC2*) promoter to drive expression of the dominant negative gai-1 protein fused to VENUS (in a *pCUC2::gai-1-VENUS* line). *CUC2* promoter drives expression in a large part of the IPR (including the boundary cells) in the SAM starting from P$_4$ in wild-type SAMs and a similar specific expression was observed in *pCUC2::gai-1-VENUS* plants (see below). The distribution of cell division angles in the entire SAM or in the IPR of *pCUC2::gai-1-VENUS* plants did not show statistically significant differences with respect to the wild-type, although unexpectedly, we found in these plants a higher frequency of 80–90° divisions in non-IPR cells (Fig. 6f–j).

The orientation of cell division has been proposed to be influenced by the geometry of the SAM and notably by tensile stresses prescribed

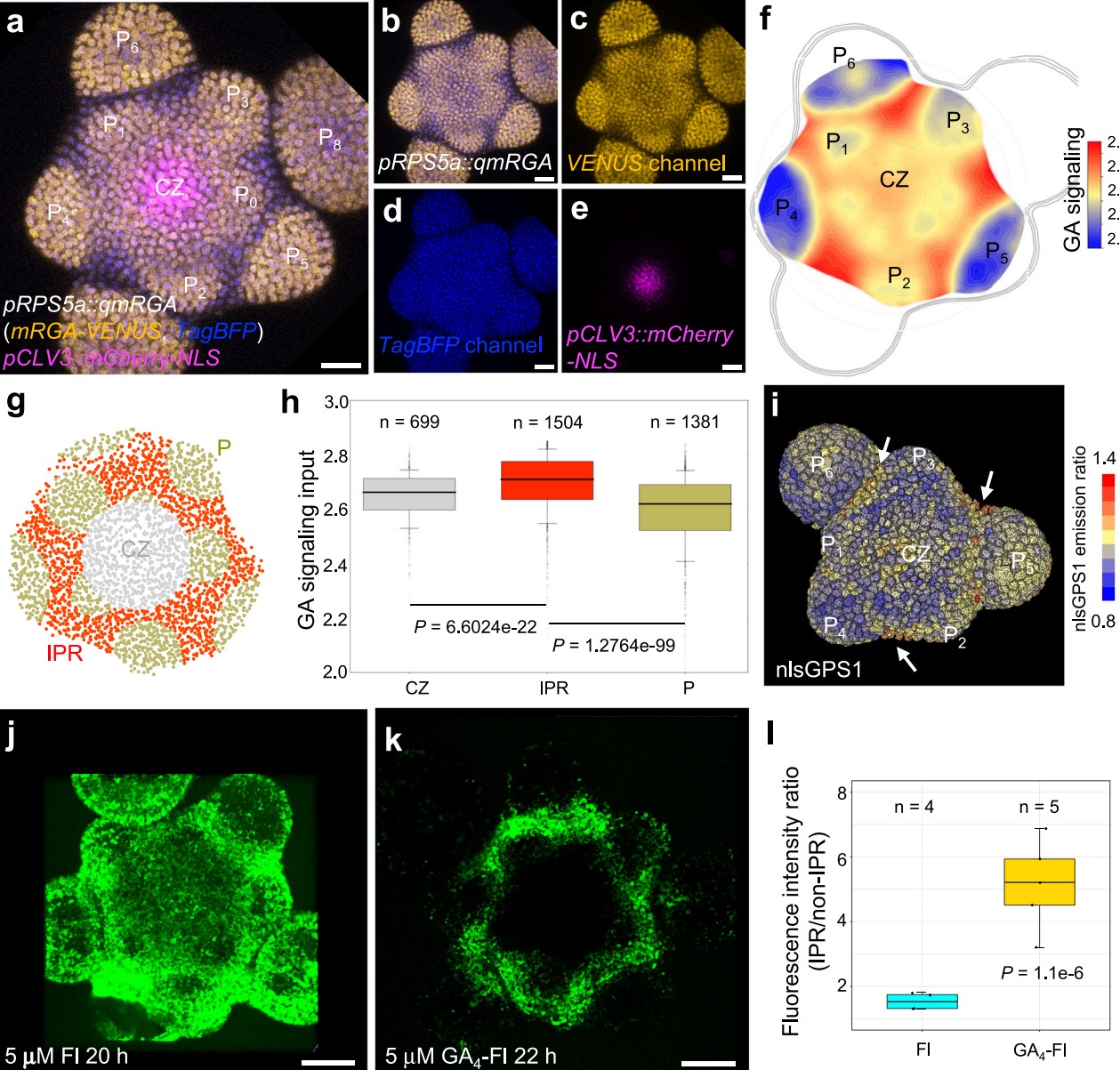

**Fig. 4 | A GA signaling activity map reveals high levels of GA signaling activity in inter-primordia cells in the SAM. a** Maximum intensity projection of three overlay channels showing the expression of *pRPS5a::qmRGA* (yellow for VENUS channel and blue for TagBFP channel) and *pCLV3::mCherry-NLS* (magenta) in the SAM. CZ, central zone; P, primordium. The subprint of P indicates the order of primordia with $P_1$ being the youngest emerged primordium. **b–e** Maximum intensity projection of two overlay channels (**b**) showing the ratio expression of *pRPS5a::qmRGA* (yellow for VENUS channel and blue for TagBFP channel) and of three individual channels (**c–e**) showing the expression of *mRGA-VENUS* (**c**), *TagBFP-NLS* (**d**) *and pCLV3::mCherry-NLS* (**e**) in the SAM, separately. The experiment was repeated twice with similar results. **f** Heatmap representation of L1 GA signaling activity averaged from seven SAMs aligned using the CLV3 domain as a reference (see supplemental methods for more details). **g, h** Quantification of GA signaling activity in central zone (CZ, $n = 699$ nuclei examined over seven independent SAM samples), inter-primordia region (IPR, $n = 1504$ nuclei examined over seven independent SAM

samples), and primordia (P, $n = 1381$ nuclei examined over seven independent SAM samples) indicated by different colors as shown in (**g**). *P*-values are from one-way ANOVA. **i** 3D visualization of nlsGPS1 emission ratio in the SAM. Primordia stages are estimated according to morphology. Arrows highlight a higher nlsGPS emission ratio in the boundaries and IPR. The experiment was repeated twice with similar results. **j, k** Fluorescence distribution in wild-type (L*er*) SAMs treated with fluorescein (Fl, (**j**)) and GA₄-Fl (**k**). **l** Comparison of the ratio of average fluorescence intensity in the IPR to that in the non-IPR (excluding primordia) between Fl ($n = 4$ independent plants) and GA₄-Fl ($n = 5$ independent plants) treatment in the SAM. *The P*-value is from one-way ANOVA with Turkey's test for multiple comparisons of means. The experiment was repeated twice with similar results. The center lines of boxes in (**h, l**) show the medians and box limits indicate the 25th and 75th percentiles. Whiskers indicate minima and maxima as determined using the R software. Scale bars = 20 μm.

by the curvature of the tissue[46]. We thus asked whether the SAM shape of the global *della* mutant and *pCUC2::gai-1-VENUS* plants was changed. As previously shown[12], the size of the global *della* mutant SAM was bigger than the wild-type (Supplementary Fig. 13a, b, d). *CLV3* and *STM* RNA in situ hybridization confirmed the enlargement of the meristem in *della* mutants, further showing a lateral enlargement of the stem cell

niche (Supplementary Fig. 13e, f, h, i). However, the SAM curvature was identical in the two genotypes (Supplementary Fig. 13k, m, n, p). We observed a comparable increase in size in the quadruple *gai-t6 rga-t2 rgl1-1 rgl2-1 della* mutant, again without modification of the curvature compared to wild-type (Supplementary Fig. 13c, d, g, j, l, o, p). The frequency of cell division orientation was also affected in the quadruple

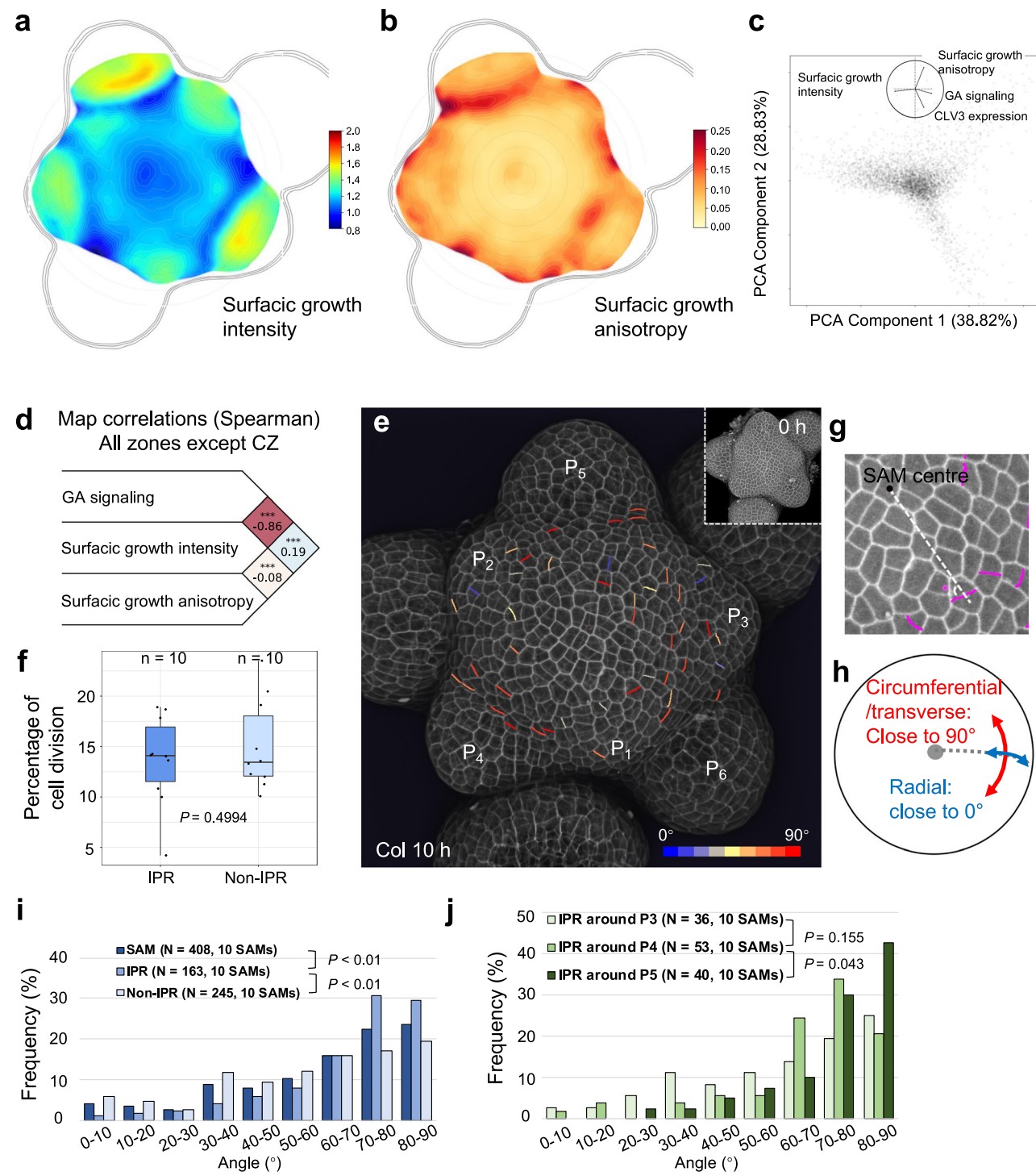

*della* mutant, but to a lesser extent than in the global *della* mutant (Supplementary Fig. 12d–f). This dosage effect, along with the absence of effects on curvature, suggests that the remaining RGL3 activity in quadruple *della* mutants limits the changes in cell division orientation caused by the loss of DELLA activity and that changes in the occurrence of transverse cell division depend on changes in GA signaling activity rather than in SAM geometry. As mentioned above, the *CUC2* promoter drives expression in the IPR in the SAM starting from P$_4$ (Supplementary Fig. 14a, b) and, by contrast, the size of *pCUC2::gai-1-VENUS* SAMs was reduced, whereas it had a much higher curvature (Supplementary Fig. 14c–h). This change in the morphology of *pCUC2::gai-1-VENUS* SAMs could generate a different mechanical stress distribution

compared to wild-type, whereby high circumferential stress starts at a shorter distance from the SAM centre[47]. Alternatively, the change in morphology in *pCUC2::gai-1-VENUS* SAMs could result from changes in regional mechanical properties induced by the expression of the transgene[48]. In both cases, this could counteract in part the effect of GA signaling changes by increasing the probability of cell division with circumferential/transverse orientation, thus explaining our observations.

Taken together, our data support a positive role for higher GA signaling in the transverse orientation of the cell division plane in the IPR. They further suggest that the curvature of the meristem can also influence the orientation of the cell division plane in the IPR.

**Fig. 5 | High GA signaling activity correlates positively with growth anisotropy and transverse cell division orientation. a, b** Averaged surface growth (**a**) and growth anisotropy (**b**) heat maps (used as proxies for cell expansion intensity and direction, respectively) in the SAM averaged from seven independent plants. **c** PCA analysis including the following variables: GA signaling, surface growth intensity, surface growth anisotropy, and CLV3 expression. PCA component 1 is mostly associated negatively with surface growth intensity and positively with GA signaling. PCA component 2 is mostly associated positively with surface growth anisotropy and negatively with CLV3 expression. Percentages are the variations explained by each component. **d** Spearman correlation analysis between GA signaling, surface growth intensity, and surface growth anisotropy at the tissue scale but excluding CZ. The numbers on the right are Spearman's rho values between two variables. Stars indicate when the correlation/anti-correlation is highly significant. **e** 3D visualization of Col-0 SAM L1 cells using confocal microscopy. New cell walls formed in the SAM (but not primordia) in 10 h are colored according to their angle values. The color bar is shown in the bottom-right corner. The insert shows the

corresponding 3D image at 0 h. The experiment was repeated twice with similar results. **f** Boxplot representation of cell division frequency in the IPR and non-IPR of Col-0 SAM ($n = 10$ independent plants). Center lines show the medians and box limits indicate the 25th and 75th percentiles. Whiskers indicate minima and maxima as determined using the R software. $P$-value is from two-sided Welch's $t$-test. **g, h** schematic diagram showing (**g**) how the angle of a new cell wall (magenta) relative to the radial direction from the center of the SAM (white dotted line) was measured (only acute angle values, i.e., 0–90°, were considered), and (**h**) the circumferential/transverse and radial orientations within the meristem. **i** Frequency histograms of division plane orientation of cells from the entire SAM (dark blue), the IPR (medium blue), and non-IPR (light blue), respectively. $P$-values are from two-sided Kolmogorov–Smirnov tests. The experiment was repeated twice with similar results. **j** Frequency histograms of division plane orientation of IPR cells around $P_3$ (light green), $P_4$ (mediate green), and $P_5$ (dark green), respectively. $P$-values are from two-sided Kolmogorov–Smirnov tests. The experiment was repeated twice with similar results.

## High GA signaling initiates internode specification in the shoot apical meristem

Transverse orientation of division planes in the IPR as a result of higher GA signaling activity opens the possibility that GAs pre-organize radial cell files in the epidermis within the SAM to specify the cellular organization later found in the internode epidermis. Indeed, such cell files are often visible in the SAM images of the global *della* mutant (Fig. 6b). Therefore, to further understand the developmental function of the GA signaling spatial pattern in the SAM, we used time-lapse imaging to analyze the cell spatial organization in the IPR in wild-type (L*er* and Col-0), global *della* mutant and *pCUC2::gai-1-VENUS* transgenic plants.

We have seen that qmRGA shows that GA signaling activity in the IPR increases from $P_1/P_2$ and peaks from $P_4$, a pattern that is consistent over time (Fig. 4a–f and Supplementary Fig. 8c–f, k). To analyze cell spatial organization when GA signaling increases in the IPR, we thus marked L*er* IPR cells above and on the side of $P_4$ according to their developmental fates analyzed 34 h after the first observation, i.e., more than two plastochrons and thus allowing to follow IPR cells during primordia development, from $P_1/P_2$ to $P_4$. We used three different colors: yellow for those incorporated into the primordia in the vicinity of $P_4$, green for those located in the IPR, and magenta for those contributing to both (Fig. 7a–c). At t0 (0 h), 1–2 layers of IPR cells were visible in front of $P_4$ (Fig. 7a). As expected, when these cells divide, they mostly do it with a transverse division plane (Fig. 7a–c). Similar results were obtained with Col-0 SAMs (focusing on $P_3$ that had a comparable folding at the boundary than $P_4$ in L*er*), although in this genotype the formation of the crease at the flower boundary hides the IPR cells more rapidly (Fig. 7g–i). Thus, the division pattern of IPR cells indeed preorganizes cells in radial files as in the internode. The organization in radial files and the localization of IPR cells in between successive organs suggest that these cells are internode precursors.

Compared to L*er*, 1-2 extra layers of IPR cells were observed in front of $P_4$ at t0 (0 h) in the SAM of global *della* mutants. These cells divided several times in 34 h (Fig. 7d–f, compared to 7a–c), and in consequence, the mostly transverse divisions of IPR cells led to a higher population of cells organized in radial cell files (Fig. 7d–f, compared to 7a–c). This indicates that the higher GA signaling activity in global *della* mutant SAMs promotes internode specification. We conducted a similar analysis in *pCUC2::gai-1-VENUS* plants. Since expression of this transgene causes changes in SAM geometry (Supplementary Fig. 12m, n, p, q), we analyzed IPR cells above the first primordium showing a comparable folding at the boundary as Col-0 $P_3$. In these plants, opposite to global *della* mutants, much fewer cell divisions occurred in the IPR and there was no clear sign of an organization in radial cell files (Fig. 7j–l), thus showing that inhibition of GA signaling in the IPR perturbs the specification of the cellular organization of internodes in the SAM. In line with these results, we were able to detect the appearance of internodes in the global *della* mutant just

below the SAM using electron microscopy, while flowers remained compacted in L*er* (Fig. 7m, n). By contrast, organs were much more compacted in the SAM of *pCUC2::gai-1-VENUS* than in Col-0 (Fig. 7o, p), consistent with the taller and shorter inflorescence stem of the global *della* mutant[44,49] and *pCUC2::gai-1-VENUS* plants, respectively (Supplementary Fig. 15). Our results thus support the hypothesis that higher GA signaling activity in the IPR specifies the cellular organization of internodes in the SAM, through a regulation of the orientation of cell division planes (Supplementary Fig. 16).

## Discussion

Here, we developed a ratiometric GA signaling biosensor, qmRGA, that provides information on GA function at the cellular level by allowing quantitative mapping of GA signaling activity that results from the combinatorial action of GA and GA receptors concentrations, with minimal interference with the endogenous signaling pathway. To this end, we have engineered a modified DELLA protein, mRGA, that has lost its capacity to bind DELLA-interacting partners but that remains sensitive to GA-induced proteolysis. qmRGA responds to both exogenous and endogenous changes in GA levels and its dynamic sensing properties allow the assessment of spatiotemporal changes in GA signaling activity during developmental processes. qmRGA is also a highly flexible tool as it can be adapted to a variety of tissues simply by changing, if needed, the promoter used for its expression, and is very likely transferable to other species given the conserved nature of the GA signaling pathway and of the PFYRE motif in angiosperms[22]. In line with this, the equivalent mutation in the rice DELLA protein SLR1 (HYY497AAA) has also been shown to inhibit SLR1 growth-repressing activity while only slightly reducing its GA-mediated degradation, which is similar to mRGA[23]. Noteworthy, recent work in *Arabidopsis* reports that a single amino acid mutation in the PFYRE domain (S474L) alters the transcriptional activity of RGA, without interfering with its ability to interact with transcription factor partners[50]. Although this mutation is in close proximity to the 3 amino acid substitutions present in mRGA, our work shows that the two mutations alter different characteristics in DELLAs. Despite that most of the transcription factor partners bind to the LHR1 and SAW domains of the DELLAs[26,51], it is possible that some conserved amino acids in the PFYRE domain contribute to stabilizing these interactions.

Internode development is a key trait for plant architecture and crop improvement. qmRGA revealed a higher GA signaling activity specifically in cells of the IPR, which are the precursors of internodes. By combining quantitative image analysis and genetics, we show that the GA signaling pattern imposes circumferential/transverse cell division planes in the SAM epidermis, shaping the cell division organization required for internode development. Few developmental regulators of the orientation of the cell division plane during development have been identified[52,53]. Our work provides a striking example

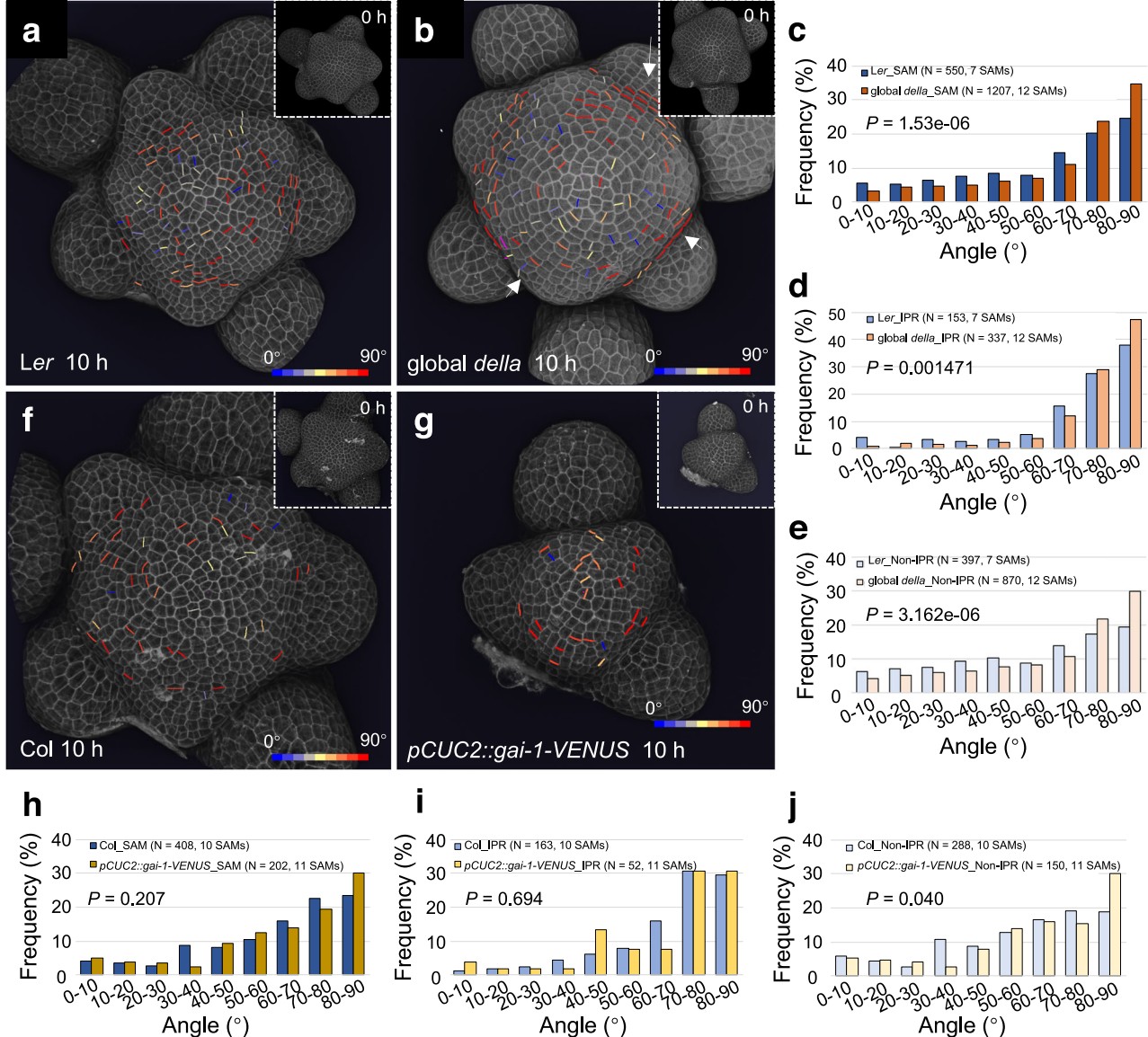

**Fig. 6 | Cell division orientation distribution in the SAM is modified in global** ***della* mutants and *pCUC2::gai-1-VENUS* transgenic plants. a, b** 3D visualization of the L1 layer of PI-stained L*er* (**a**) and global *della* mutant (**b**) SAM using confocal microscopy. New cell walls formed in the SAM (but not primordia) in 10 h are shown and colored according to their angle values. Inserts show the SAM at 0 h. The color bars are shown in the bottom-right corners. Arrows in (**b**) highlight examples of aligned cell files in global *della* mutant. The experiment was repeated twice with similar results. **c–e** Comparison of the frequency distribution of division plane orientation of cells in the entire SAM (**d**), IPR (**e**), and non-IPR (**f**) between L*er* and global *della*. *P*-values are from two-sided Kolmogorov–Smirnov tests. **f, g** 3D visualization of confocal image stacks of PI-stained SAMs of Col-0 (**i**) and *pCUC2::gai-1-VENUS* (**j**) transgenic plants. New cell walls formed in the SAM (but not primordia) in 10 h are shown as in (**a, b**). The experiment was repeated twice with similar results. **h–j** Comparison of the frequency distribution of division plane orientation of cells located in the entire SAM (**h**), IPR (**i**), and non-IPR (**j**) between Col-0 and *pCUC2::gai-1-VENUS* plants. *P*-values are from two-sided Kolmogorov–Smirnov tests.

where GA signaling activity regulates this cellular parameter. DELLA can interact with the prefoldin complex[41] and GA signaling could thus regulate the orientation of the cell division plane through a direct effect on cortical microtubule orientation[40,41,54,55]. The fact that we show that unexpectedly not cell elongation nor cell division but only growth anisotropy correlates in the SAM with higher GA signaling activity is coherent with a direct effect of GAs on cell division orientation in the IPR. However, we cannot eliminate the possibility that this effect could also be indirect, e.g. mediated by GA-induced softening of the cell wall[56]. Changes in cell wall properties induce mechanical stress[57,58] that could also affect cell division plane orientation by acting on cortical microtubule orientation[39,46,59]. A combined effect of GA-induced mechanical stress and direct regulation by GA of microtubule orientation could then participate in creating the specific pattern of

cell division orientation in the IPR to specify the internode and further work is needed to test this idea. Likewise, previous works have highlighted the importance of the DELLA-interacting proteins TCP14 and 15 in the control of internode patterning[60,61] and these factors could convey GA action, together with *BREVIPEDICELLUS* (*BP*) and *PENNYWISE* (*PNY*), which regulate internode development and have been shown to affect GA signaling[2,62]. Given the fact that DELLA interacts with the signaling pathways of brassinosteroids, ethylene, jasmonic acid, and ABA[63,64] and that these hormones can influence microtubule orientation[65], the effects of GA on cell division orientation could also be mediated together with other hormones.

Early cytological studies showed that both the inner tissues and the peripheral zone of the SAM are required for internode development in *Arabidopsis*[2,42]. The fact that GAs positively regulate cell

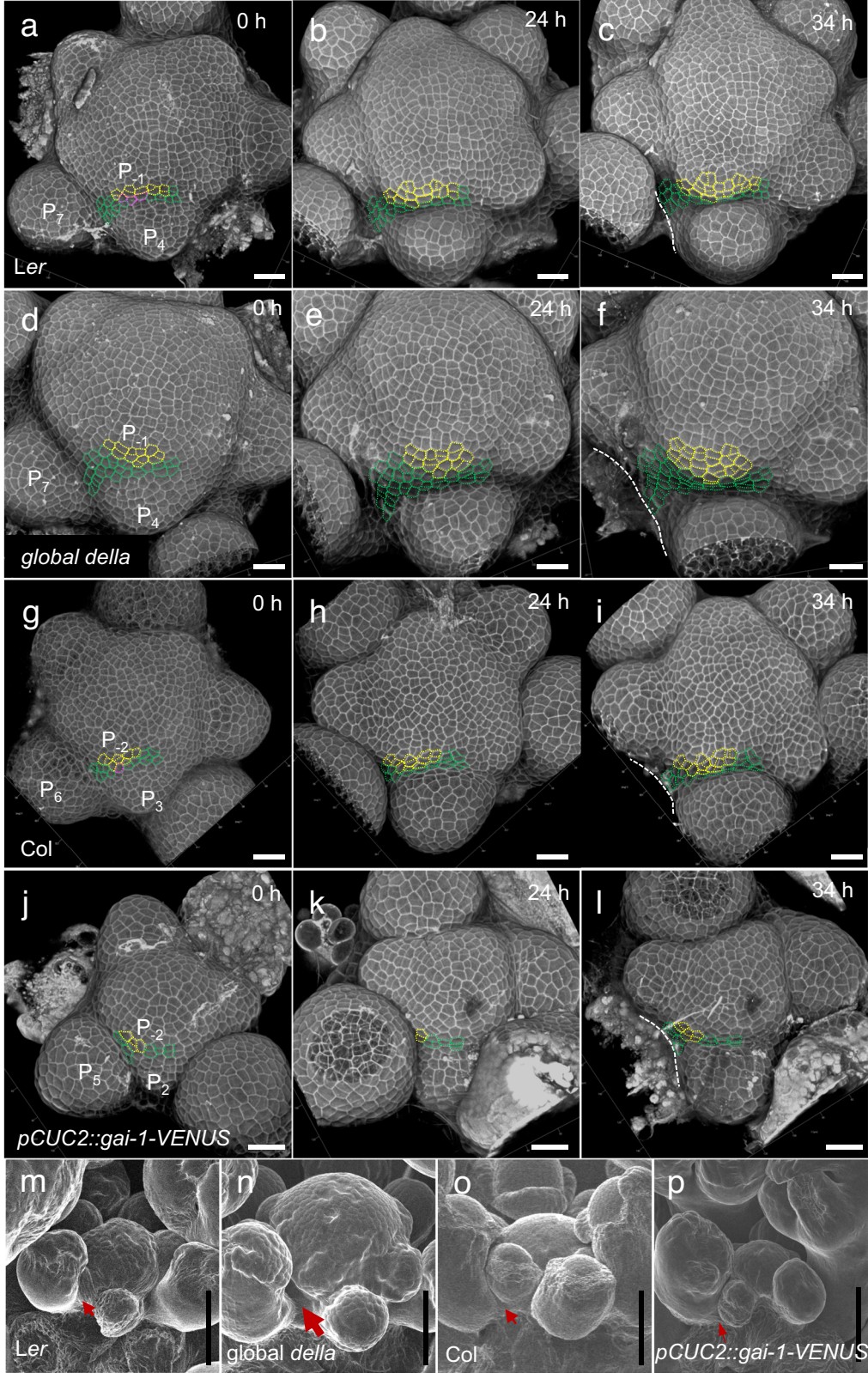

division in the inner tissues[12] supports a dual function of GAs in regulating meristem size and internode at the SAM. Patterns of oriented cell division are also highly regulated in the inner SAM tissues, and this regulation is essential to stem growth[52]. It will be interesting to explore whether GAs also play a role in orienting cell division planes in the inner tissues of the SAM and thus synchronize internode specification and development within the SAM.

## Methods

### Growth conditions and plant material

Plants were grown on soil or in vitro on 1x Murashige-Skoog (MS) medium (Duchefa) supplemented with 1% sucrose and 1% agar (Sigma) under standard conditions (16 h photoperiod at 22 °C), except for hypocotyl and root growth experiments, for which the seedlings were grown on vertical plates under continuous light and 22 °C. For

**Fig. 7 | Emergence of the internode cellular organization in the inter-primordia region in wild-type and in plants with modified GA signaling activity. a–l** Time-lapse (0 h, 24 h, 34 h) visualization of the L1 of the SAM from confocal microscopy of L*er* ((**a**–**c**) *n* = 4 independent plants), global *della* ((**d**–**f**) *n* = 5 independent plants), Col-0 ((**g**–**i**) *n* = 3 independent plants) and *pCUC2::gai-1-VENUS* transgenic ((**j**–**l**) *n* = 4 independent plants). The imaging time is indicated at the right-up corner of each panel. Cells outlined with yellow dotted lines are those incorporated into a primordium at 34 h, and cells in green are found in the IPR at 34 h. Those in

magenta produce both primordium and IPR cells. White dotted lines mark the edge of the elder primordium (that have been removed at 34 h except in (**c**)). **m**–**p** Scanning electron microscopy images of the shoot apex of L*er* (**m**), global *della* (**n**), Col-0 (**o**), and *pCUC2::gai-1-VENUS* (**p**) allowing to analyze the early establishment of internode just below the SAM. The different size of arrows indicates differences in early internode length in the different genetic backgrounds. The experiment was repeated twice with similar results. Scale bars = 20 μm (**a**–**l**), 50 μm (**m**–**p**).

experiments with nitrate, plants were grown on MS-modified medium without nitrogen (bioWORLD plant media) supplemented with an adequate nitrate concentration (0 or 10 mM $KNO_3$), 0.5 mM $NH_4$-succinate, 1% sucrose and 1% type-A agar (Sigma) under long-day photoperiod.

The mutant and transgenic lines we used and all the plasmids and transgenic lines generated for this paper are listed in Supplementary Table 2. The plasmid construction and plant transformation were conducted as below. RGA cDNA (AGI code AT2G01570) and RGA^m1, RGA^m2, RGA^m3, and RGA^m4 mutant variants were obtained by PCR using specific primers numbered 1 to 8 in Supplementary Table 3, and inserted into pDONR221 (Thermo Fisher Scientific) by Gateway cloning and recombined with pB7WGF2[66] to generate *p35S::RGA-GFP* and *p35S::RGA^m1/m2/m3/m4-GFP*. To generate *pRPS5a::qRGA, pUB-Q10::qRGA, pRPS5a::qmRGA, pUBQ10::qmRGA, pRPS5a::RGA^m1/m3/m4-VENUS-2A-TagBFP, pRPS5a::VENUS-2A-TagBFP, pUBQ10::RGA^m1/m3/m4-VENUS-2A-TagBFP,* and *pCUC2::gai-1-VENUS*, the promoter region of *pRPS5a* (1.7 kb fragment), *pUBQ10* (2.5 kb fragment) or *pCUC2* (3.2 kb fragment) inserted into pDONR P4-P1R (Thermo Fisher Scientific), *RGA-VENUS, mRGA-VENUS, RGA^m1/m3/m4-VENUS, VENUS-N7* (for *pRPS5a::VENUS-2A-TagBFP*) or *gai-1* cDNA inserted into pDONR221, and *2A-TagBFP-SV40nls* or *VENUS* (for *pCUC2::gai-1-VENUS*) inserted into pDONR P2R-P3 (Thermo Fisher Scientific), were recombined into pB7m34GW[66].

d17RGA (RGA deleted of the 17 amino acids DELLAVLGYKVRSSEMA composing the DELLA domain[33]) and d17mRGA mutant variant obtained by PCR using primers 1, 2, 5, and 6 (Supplementary Table 3) were inserted into pDONR221 and recombined into pB7m34GW with *p35S* (inserted into pDONR P4-P1R) and *VENUS* (inserted into pDONR P2R-P3) to generate *p35S::d17RGA-VENUS* and *p35S::d17mRGA-VENUS*. *d17RGA-VENUS* and *d17mRGA-VENUS* were then amplified by PCR using primers 1 and 12, inserted into pDONR221 and recombined as previously with *pRPS5a* or *pUBQ10* and *2A-TagBFP-SV40nls* to generate *pRPS5a::qd17RGA, pUBQ10:qd17RGA, pRPS5a::qd17mRGA, pUBQ10:qd17mRGA*.

To obtain *pUBQ10::GID1a-mCherry*, *GID1a* cDNA inserted into pDONR221 was recombined with pDONR P4-P1R-*pUBQ10* and pDONR P2R-P3-*mCherry* into pB7m34GW. *p35S:IDD2-RFP* was obtained by recombining *IDD2* cDNA inserted in pDONR221 into pB7RWG2[66]. To get *pGID1b::2xmTQ2-GID1b*, a 3.9-kb fragment upstream of the coding region of *GID1b* and a 4.7-kb fragment including *GID1b* cDNA (1.3 kb) and terminator (3.4 kb) were first amplified using primers in Supplementary Table 3, then inserted into pDONR P4-P1R (Thermo Fisher Scientific) and pDONR P2R-P3 (Thermo Fisher Scientific), respectively, and finally recombined into pGreen 0125[67] destination vector with pDONR221 2xmTQ2[68] by Gateway cloning. To make *pCUC2::LSSmOrange*, the promoter sequence of *CUC2* (3229 bp upstream the ATG), followed by the coding sequence for the large stokes shift mOrange (LSSmOrange)[69] with an N7 nuclear localization signal, and the NOS transcription terminator, was assembled into the pGreen Kanamycin destination vector using 3 fragments Gateway recombination system (Invitrogen). Plant binary vectors were inserted into *Agrobacterium tumefaciens* GV3101 strain and respectively introduced into *Nicotiana benthamiana* leaves by agro-infiltration and *Arabidopsis* Col-0 by floral dip. *pUBQ10::qmRGA pUBQ10::GID1a-mCherry* and *pCLV3::mCherry-NLS qmRGA* were isolated from F3 and F1 progeny of the appropriate crosses, respectively.

## Pharmacological treatments
Chemical treatments with GA ($GA_3$, Sigma, or Duchefa) and paclobutrazol (Pac, Duchefa) were performed at the concentrations and time indicated in the figures. For Y2H assays, 100 μM $GA_3$ was added in selective media to promote interaction between RGA and GID1. For RGA and RGA^m1/m2/m3/m4 degradation kinetics, *N. benthamiana* agro-infiltrated leaf discs were incubated with 100 μM $GA_3$ and 100 mM cycloheximide (Sigma) over 300 min. For hypocotyl length measurements and fluorescence analyses of qmRGA hypocotyls and roots, seedlings were grown for 5 days on MS agar medium and then transferred for 4 days on MS agar plates supplemented with 5 μM Pac or 10 μM $GA_3$. For $GA_3$ treatment on qmRGA, $GA_{4+7}$ treatment on nlsGPS1, and GA-Fl treatment on L*er*, dissected shoot apices were cultured into Apex Culture Medium (ACM, 1/2x MS medium (Duchefa), 1% sucrose, 1% agarose, 2 mM MES (Sigma), 1x vitamin solution (myo-inositol 100 mg/L, nicotinic acid 1 mg/L, pyridoxine hydrochloride 1 mg/L, thiamine hydrochloride 10 mg/L, glycine 2 mg/L), 200 nM N6-benzy-ladenine, pH 5.8) with indicated concentrations of GA/GA-Fl, and also immersed under 200 μL of GA/GA-Fl solution of indicated concentrations for a indicated period of time. For Pac treatment on qmRGA, 50 μM Pac (dissolved in ethanol and diluted in water) was sprayed on the whole inflorescence every two days for a period of 5 days before observation.

## Yeast two-hybrid assays
For Y2H assays, both the full-length and the C-terminal part of RGA and RGA^m1/m2/m3/m4 (named M5 version, amino acids 199 to 587; the N-terminal part is subject to self-activation in yeast[70]) inserted into pDONR207 were recombined into pGBKT7 (Clontech) to obtain BD-RGA, BD-RGA^m1/m2/m3/m4, BD-M5RGA and BD-M5RGA^m1/m2/m3/m4. On the other hand, JAZ1, TCP14, IDD2, BZR1, GID1a, GID1b and GID1c cDNAs inserted into pDONR221 were fused to the activation domain GAL4 (AD) after recombination into pGADT7 (Clontech).

Direct interaction assays were carried out following the Clontech procedures. BD-M5DELLA and AD-JAZ1, AD-TCP14, AD-IDD2, AD-BZR1 and, on the other hand, BD-DELLA (full-length) and AD-GID1 constructs were co-transformed in the yeast strain AH109 and interactions tests were surveyed on selective medium lacking tryptophan, leucine, and histidine. In some cases, the medium was supplemented with 3-amino-1, 2, and 4 triazole (3AT, Sigma) or with GA to promote interaction between DELLA and GID1.

To confirm that the RGA^m2 mutation impaired interaction with all RGA protein partners, we used the C-terminal part of RGA and RGA^m2 (amino acids 199 to 587), to probe the Arabidopsis transcription factors REGIA + REGULATORS (RR) arrayed library, following the protocol described in Castrillo et al.[71] TFs in the RR library were fused to GAL4 activation domain of the pDEST22 vector and independently transformed into the yeast strain YM4271 in 96-well plates. The RGA^m2 protein fused to the GAL4 BD in the pDEST32 vector was transformed into the pJ694 yeast strain and used as bait. Replicates of the library were grown overnight on SD-Trp solid media and inoculated together with 100 μL of an overnight RGA^m2 culture grown on SD-Leu on microtiter plates containing 100 μL of YPDA per well. Plates were incubated for 2 days at 30 °C for mating, and diploid colonies were selected in new 96-well plates containing 200 μL of SD-Leu/Trp. 5 μL of the diploid cell cultures were lastly tested for protein

interaction by placing them on solid SD medium lacking both Leu and Trp (positive growth control), and on SD medium lacking Leu, Trp and His, in the presence of 1 mM 3-aminotriazol (3-AT) (Sigma-Aldrich). Results were expressed in the form of a heat map for the strength of interaction according to the colony growth after five days of incubation at 30 °C.

## Co-immunoprecipitation assays

CoIP assays were performed on *N. benthamiana* agro-infiltrated leaves with *p35S::IDD2-RFP*, *p35S::TCP14-RFP*, *p35S::RGA-GFP* or *p35S::mRGA-GFP*. Three days after infiltration, total proteins were extracted with the native extraction buffer [Tris-HCl (pH 7.5) 50 mM, glycerol 10%, non-idet P-40 0.1% supplemented with Complete Protease Inhibitors 1X (Roche)], and then incubated for 2 h at 4 °C with 50 μL of anti-GFP antibody conjugated with paramagnetic beads (Miltenyi Biotec, Catalog number 130-091-125). After incubation, samples were loaded onto a magnetic column system (μ columns; Miltenyi Biotec) to recover the immunoprotein complexes. The immunoprecipitated (RGA-GFP and mRGA-GFP) and co-immunoprecipitated (IDD2-RFP and TCP14-RFP) proteins were detected by western-blot with a 2000-fold dilution of anti-GFP (JL8; Clontech, Catalog number 632380) and anti-RFP (6G6 α-Red, Chromotek, Catalog number 51020014AB), respectively.

## Immunodetection analyses

Plant materials were ground in the protein extraction buffer (Tris-base 62,5 mM pH6.8; urea 4 M; sodium dodecyl sulfate 3% (p/v); dithio-threitol (DTT); glycerol 10% (v/v), bromophenol blue 0,1% (p/v)), followed by heating at 95 °C for 5 min. After centrifugation at 13000 *g* for 5 min, total proteins were separated on 8.5% SDS-PAGE gel and transferred to an immobilon-P (PVDF) membrane (Millipore). Membranes are then saturated with blocking buffer (TBS 1x, tween-20 0.1%; milk 5%) and incubated with a 2000-fold dilution of anti-GFP (JL8; Clontech, Catalog number 632380), anti-RFP (6G6 α-Red, Chromotek, Catalog number 51020014AB), anti-RGA (Agrisera, Catalog number AS111630), anti-HA (Sigma, Catalog number H9658, clone HA-7) or anti-actin (Agrisera, Catalog number AS 132640) antibodies, and a 5000-fold dilution of peroxidase-conjugated goat anti-rabbit or mouse IgG (Invitrogen, Catalog number G21040 and G21234, respectively). Signals were detected using the Luminata Forte Western HRP Substrate (Millipore).

## Transactivation assays

Transactivation assays were performed using the Dual-Glo Luciferase Assay System (Promega). *N. benthamiana* leaves were first agro-infiltrated with *pTCP::LUC* and *pBRE::LUC* reporter constructs[66], *p35S::3xHA-VP16-TCP14* or *p35S::3xHA-VP16-BZR1* encoding effector proteins[66], and *p35S::RGA-GFP* or *p35S::mRGA-GFP*. Three days after infiltration, total proteins were extracted in lysis buffer (Promega), then Firefly and control Renilla LUC activities were quantified with FLUOstar Omega luminometer (BMG Labtech) using OMEGA2 software version 5.50 R4. For loading control, protein levels were analyzed by immunodetection.

## GUS staining

For GUS expression detection in inflorescence apices, 28-day-old plants grown under long-day (LD) conditions were used. After fixation in 90% cold acetone at room temperature for 20 min, shoot apices were transferred into a GUS staining solution containing 1 mM potassium ferrocyanide, 1 mM potassium ferricyanide, and 1 mg/mL 5-bromo-4-chloro-3-indolyl-*β*-D-glucuronide (X-Gluc), vacuum-infiltrated on ice for 15 min, and incubated overnight at 37 °C in dark before washing with ethanol series and microscope detection. For the experiment with seedlings, 7-day-old plantlets grown in vitro under LD conditions were vacuum-infiltrated for 15 min in GUS staining solution containing 2 mM potassium ferrocyanide, 2 mM potassium ferricyanide and 0.25 mg/mL X-Gluc, and incubated at 37 °C for 24 h.

Then GUS solution was replaced with ethanol 70% and seedlings were observed using an optical microscope.

## In situ hybridizations

RNA in situ hybridizations were performed on shoot apices with ~1 cm stem[72] that were collected and immediately fixed in FAA solution (3.7% formaldehyde, 5% acetic acid, 50% ethanol) precooled at 4 °C. After vacuum treatments for 2 × 15 min, the fixative was changed and the samples were incubated overnight. Antisense probes of the cDNA and 3′-UTR of *GID1a*, *GID1b*, *GID1c*, *GAI*, *RGL1*, *RGL2*, and *RGL3* were synthesized as described in Rozier et al.[73] using primers indicated in Supplementary Table 3. Immunodetection of the digoxigenin-labeled probes was performed using an anti-digoxigenin antibody (3000-fold dilution; Roche, Catalog number: 11 093 274 910), and sections were stained with 5-bromo-4-chloro-3-indolyl phosphate (BCIP, 250-fold dilution)/nitroblue tetrazolium (NBT, 200-fold dilution) solutions.

## Microscopy

For confocal microscopy observations of hypocotyl and root, complete seedlings were put on slides, and images were obtained with a Zeiss LSM 780 confocal microscope. For SAM live imaging, dissected SAMs were let to recover overnight after dissection. Confocal images were then taken with a Zeiss LSM 710 or 700 confocal laser scanning microscope equipped with a water-dipping lens (W Plan-Apochromat 40x/1.0 DIC). To observe GFP or GFP with propidium iodide (PI), a laser of 488 nm was used to excite, and the emission for GFP was 500–520 nm (in some cases it was 500–540 nm), and 610–650 nm for PI. For imaging of qmRGA or qmRGA with mCherry and PI, lasers of 514 nm, 405 nm, 561 nm, and 488 nm were used for the excitation of VENUS, TagBFP, mCherry, and PI, respectively, and the corresponding emission wavelengths were 520–560 nm, 430–460 nm, 580–615 nm and 620–660 nm. mTURQUOISE2 (mTQ2) was excited by a 445 nm laser and the emission range was 470–510 nm. For FRET detection of nlsGPS1, an acquisition mode of spectral imaging (λ-scan) with emission wavelength from 461 nm to 597 nm was used with excitation at 458 nm.

For optical microscopy, photographs of plants were taken with a LEICA MZ12 stereoscopic microscope equipped with a ZEISS AxioCam ICc5 camera head or Canon camera.

For scanning electronic microscopy of the fresh shoot apex, a Hirox SH-3000 table-top microscope equipped with Coolstage (−20 °C to −30 °C) was used.

## Image processing

Confocal stacks were processed in Fiji (fiji.sc) to get max projection or orthogonal views. Older primordia were also removed in Fiji. For 3D visualization of confocal stacks, we used the Zeiss ZEN2 software (Fig. 7) and a rendering using the VTK library[74] (Figs. 5e and 6). In both cases, parameters were adjusted accordingly to show mainly the L1 cells.

## Quantification and statistical analysis

**Fluorescence quantification.** For qmRGA quantification and visualization, images of hypocotyls and roots were analyzed using a Python script that performs nuclei detection and signal quantification in 3D. The details of the algorithms used in the analysis are provided in Supplementary Methods. The quantification of fluorescence ratios in the control lines was performed using the same computational pipeline. A modification was done to this pipeline to be able to compare quantitatively fluorescence ratio distributions between qmRGA and the *pRPS5a::VENUS-2A-TagBFP* control line (Supplementary Fig. 8). To transform the distributions onto comparable value ranges, we standardized the values of ratios obtained for each image by dividing them by their mean, in order to always have ratio distributions centered on 1.

For nlsGPS1 quantification and visualization, a series of Fiji macros based on a published plugin "3D ImageJ suite" were used[75]. Briefly, a "seg-auto.ijm" macro was run for the segmentation of nuclei based on the sum of signals from all 14 channels of the spectral imaging. After removing little objects corresponding to signal noise using a "object-screening.ijm" macro, the ratio of DxAm (channel 8) to DxDm (channel 3) of each segmented nucleus was calculated by running a "3D-ratio.ijm", and from here, the 3D construction of the SAM, with every nucleus showing its ratio value, was also achieved and printed. Statistical analysis was done in R.

**SAM image sequence analysis.** To analyze time-lapse confocal acquisitions of SAMs and obtain quantitative measures of GA signaling as well as cellular growth parameters, we developed a computational pipeline. It consists of 3D watershed segmentation for cell identification from the PI staining, nuclei detection and qmRGA quantification, temporal registration using the PI image channel, and surface growth estimation based on expert cell lineages. Individual SAM sequences were then aligned in order to perform population-scale statistics. Extensive details on the algorithms used in this pipeline are provided in Supplementary Methods.

**Hypocotyl length measurement.** To measure hypocotyl length, seedlings were grown in vertical MS medium (Duchefa) with 0.8% (w/v) phytoagar and 1% sucrose for 5 days and transferred to MS, MS supplemented with 10 μM GA$_3$ or 5 μM Pac for 4 days in continuous light at 22 °C. Plates were scanned and hypocotyl length was measured from the images using Fiji.

**Cell division orientation quantification.** New cell walls were identified by comparing images obtained at 0 h and 10 h, and masked with a manually drawn line in Fiji. Then, a macro was used to skeletonize and then to measure the angles of the drawn cell walls with an expert-defined center of the SAM. Further angle frequency distribution was done in Excel using the Pivot table. Statistical analyses of Kolmogorov–Smirnov tests were performed using an online tool (http://www.physics.csbsju.edu/stats/KS-test.n.plot_form.html) or in R.

**Curvature quantification.** The MorphoGraphX software was used to quantify the curvature of L1 cells of SAMs of different genetic backgrounds. Statistical analyses were done in R.

**Meristem size measurement.** To measure meristem size, the SAM radius was determined by drawing a circle that covers I$_1$ and I$_2$ and so that the center of the center is roughly overlapping with the geometrical center of the SAM surface using Fiji. Statistical analysis was done in R.

**Synthesis and characterization of GA-fluorescein (GA-Fl)**
Extensive details on the synthesis and characterization of GA-fluorescein (GA-Fl)[19] are provided in Supplementary Methods.

**Reporting summary**
Further information on research design is available in the Nature Portfolio Reporting Summary linked to this article.

## Data availability
All data generated and analyzed during this study are either included in this published article (and its supplementary information files) or available for download from *Zenodo* (https://doi.org/10.5281/zenodo.10934411). Source data are provided in this paper.

## Code availability
Quantitative image and geometry analysis algorithms are provided in Python libraries timagetk, cellcomplex, tissue_nukem_3d, and sam_atlas (https://gitlab.inria.fr/mosaic/) made publicly available under the CECILL-C license. The script used to process hypocotyl images is publicly available in the qrga_nuclei_quantification project (https://gitlab.inria.fr/gcerutti/). The pipelines used to analyze SAM image sequences and to produce map visualizations are provided in a separate project (https://gitlab.inria.fr/mosaic/publications/sam_spaghetti) as Python scripts.

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

## Acknowledgements

We thank the colleagues of the Laboratoire Reproduction et Développement des Plantes and the Institut de Biologie Moléculaire des Plantes for fruitful discussions on this work and for sharing material and advice. We thank Mathilde Sirlin-Josserand for her help with graphical representation and statistics in Figs. 1f, g, and 2c, and in Supplementary Fig. 2g. We also thank Miguel Perez-Amador, Jan Lohmann, Yvon Jaillais, Tai-ping Sun, Miguel Blazquez, David Alabadi, and the NASC for seeds and plasmids. We thank the personnel of SFR Biosciences (UMS3444/CNRS, US8/Inserm, ENS de Lyon, UCBL) facility PLATIM, and especially Jacques Brocard, for assistance with microscopy and image analysis. We also thank Claire Lionnet for her help with image analysis. This work was supported by the ANR-16-CE13-0014 (GrowthDynamics) grant to T.V. and P.A.; a European Research Council grant (GAtransport) to R.W.; a Guangdong Laboratory for Lingnan Modern Agriculture Grant NG2021001 to B.S.; a grant from the Israel Science Foundation (grant no. 1057/21) to R.W.

## Author contributions

P.A. and T.V. designed the study and supervised the work; B.S., A.F.-B., P.A., and T.V. designed the experiments. B.S., A.W., and A.M.J. performed the FRET analysis; A.N.-G. and S.P. conducted the large-scale two-hybrid analysis; S.L. and R.W. designed the synthesis strategy and synthesized the GA-Fl fluorescent probes. G.C. developed the pipeline for quantitative image analysis with the help of J.L., B.S., A.F.-B., J.M., and G.C. performed the image analysis. B.S., A.F.-B., C.G.-A., L.J., G.B., E.V., L.S.-A., and J.-M.D. performed all the other experiments; all authors were involved in data analysis; B.S., A.F.-B., P.A., and T.V. wrote the manuscript with inputs from all authors.

## Competing interests

The authors declare no competing interests.
