## [Peer Review File · Nature Communications]

A quantitative gibberellin signaling biosensor reveals a role for gibberellins in internode specification at the shoot apical meristemReviewer #1 (Remarks to the Author):

In this manuscript, the authors developed a GA signaling biosensor by engineering one of the DELLA proteins, try to suppress its master regulatory function in GA transcriptional responses while preserving its degradation upon GA sensing. Using this reporter line, they detected GA distribution in several tissues, such as hypocotyl and shoot apical meristem and get some information about GA spatial role in tissue development. They also used some GA signaling mutants to test their hypothesis, but genetic evidence is not sufficient. There are many points that should be carefully checked before the conclusion was drawn.

Major points:

1. The construction of reporter line is not reliable. First, the choose of one of the DELLA proteins is not sufficiently evidenced. Peptides among DELLA proteins and the structure of whole protein is related to both its two functions, three residues mutation is not convinced. Second, this reporter depends on the degradation of DELLA upon GA sensing, but there are many intermediate steps from GA level to its function. The distribution of receptors, catalytic enzymes and proteasome pathway are all involved. So this reporter line is not a good system to reflect GA signal in vivo.
2. In plant, GA level is low and the span is narrow. Plant response to hormone is quite sensitive. The exogenous fluorescent protein GFP/BFP may be not sensitive enough to reflect define GA level.
3. GA concentration and signal should be distinguished.
4. The concentration of GA (100 μ m) in exogenous treatment is too high.
5. How about qRGAmPFYR signal in the knock-out mutants GA receptor?
6. WUS and CLV3 should be detected in della mutants to confirm developmental defect in shoot apical meristem.
7. How do GID1a,b,c distribute in L1-3 layers in SAM?
8. The finding that GA does not regulate cell expansion or division but play a role in cell plane orientation is not consistent with previous study. How about GA distribution in cells? Is the change in cell plane orientation caused by unbalance of GA?
9. To explore the contribution of GA signalling to cell division, genetic evidence is missed.
10. The influence of GA to cell division is direct or indirect? How about other hormones. Whether GA play a role through other hormones? Is this function of GA similar in other tissues except SAM? Do the GA biosynthetic mutants have similar phenotypes in cell division or it is just accessory role of GA downstream target genes?
11. GAI, RGA, RGL GUS lines only show the promoter activity, protein fusion lines should be used instead.
12. In the construction of pCUC2::*gai-1*-VENUS, why chose CUC2? It is not the member with the highest activity in CUC family.
13. Cell division angles are not changes in pCUC2::*gai-1*-VENUS , but frequency is higher, and taken together, the phenotype of SAM in GA mutants implies that GA dose not have direct function on SAM but more likely through other hormones such auxin and cytokinin.
14. P15, L343-344, "the curvature of the tissue" is phenotype, but not the cause.
15. Figure7, in the genetic experiment, biosynthetic mutants should be used instead of della mutant.

Minor points:

1. Figure 1, BRE was proved to be a promoter element that bound by BZR1 and repress gene expression downstream BR signaling pathway (Zhiyong Wang, Science, 2005). Here, the results showed activation of target genes, it is conflict with previous reports.
2. P7, L152, "than" should be changed to "to".
3. Typo error of "signaling" all over the manuscript.

Reviewer #2 (Remarks to the Author):

In the manuscript by Shi et al. entitled "A quantitative gibberellin signalling biosensor reveals a role for gibberellins in internode specification at the shoot apical meristem", the authors generated a degradation-based GA signaling reporter using a mutated/modified RGA DELLA protein GA co-receptor and investigated GA signaling in the shoot apical meristem (SAM).

First the authors generated four RGA mutant versions and tested them in yeast-2-hybrid assays for disrupted interaction with known interaction partners, but intact interaction with GID1 GA receptors, and for intact GA-induced degradation. Based on their results, RGAm2 was chosen for further characterization. Yeast-2-Hybrid analyses indicated a strong impairment of RGAm2 interaction with RGA interaction partners. Using a transient tobacco infiltration assay it was confirmed that RGAm2 had no effect on BZR1 and TCP transcriptional response in contrast to RGA. Co-IP data of RGA(m2) interaction with IDD2 (Figure 1e) were not convincing. Based on the above results, the authors generated pUBQ10: or pRPS5a:RGAm2(renamed to RGAmPFYR)-Venus-2A-TagBFP-NLS reporter constructs (in short qRGAmPFYR; I will use here qRGAm2 for simplicity) and generated transgenic Arabidopsis lines. In general, the authors show that qRGAm2 lines are not disturbed in GA responses and respond to GA by VENUS protein level or fluorescence decrease, although not complete datasets for both promoter:qRGAm2 lines are provided (see Fig. 2).

Then the authors analyzed GID1a-c GA-receptor expression patterns in Arabidopsis. The data indicated distinct, but overlapping expression patterns, although the data were limited to analyses in the SAM and in 7-day-old seedlings. More importantly, the authors provided data from hypocotyls, indicating that overexpression of the GA receptor GID1a lead to an enhanced GA signaling reported by qRGAm2. Furthermore, qd17RGAm2 appeared insensitive to GA in hypocotyls. This line may provide an important negative control, which in theory would be representative for the qRGAm2 GA signaling minimum GA-independent reporter level. qRGAm2 (strange image labeling in Fig. 4a) analyses in the SAM indicated that GA signaling decreased during the development of flower primordia (older primordia exhibited lower GA signaling), while GA signaling was high in inter primordia regions (IPR). Complementary analyses using the GA FRET-based Biosensor GPS1 and fluorescent GSs indicated that GAs accumulate in IPR, but not in the central zone (CZ) in which GA signaling (qRGAm2) also appeared relatively high. In these analyses it would have been good if the authors would have conducted parallel analyses with their important negative control reporter qd17RGAm2.

Then the authors compared GA signaling in the SAM with cellular growth in the SAM. These rather complex analyses were at least for me quite difficult to understand given the limited explanation in the main text and the figure caption of Fig. 5. It appeared an anticorrelation of GA signaling and cell expansion, and a low correlation between GA signaling and growth anisotropy. From the data the authors concluded that higher GA signaling in the IPR acts on cell growth orientation rather than expansion. From their data the authors concluded that GA signaling controls the orientation of cell division. This hypothesis was supported by analyses using a global della mutant in which GA signaling is strongly increased. Conversely, suppression of GA signaling in the IPR via pCUC2::gai-1-Venus expression rather affected division plane angles in non-IPR cells and the curvature of the SAM. In this respect the authors should clearly check via fluorescence analyses in the SAM if the pCUC2::gai-1-Venus construct is properly expressed in the SAMs CUC2 domain.

Finally, the authors showed some SAM data to indicate the influence of GA signaling (via genetic manipulation) on internode organization. As I am not an expert of SAM, I was not able to follow these explanations and presented data. Also, I suggest that the authors conduct similar timelapse SAM analyses (as in Fig. 7) using their qRGAm2 and qd17RGAm2 reporters to support their claims, especially given the title of the manuscript. Adding the global della mutant to Extended Data Fig. 14 would also be helpful.

Other comments:

Lines 98-99) Define the exact mutations that were introduced in RGAm1 to RGAm4, as this is not shown in Figure 1 and Extended Data Figure 1) or define them in the respective figures.

Figure 1a and extended Data Figure 1a) Indicated mutated amino acid numbers.

Figure 1e) In the co-IP experiment RGA-GFP with IDD2-RFP, the positive control is barely detectable. How can the authors conclude from such data that the RGAm2-GFP interaction with IDD2-RFP is abolished when not even the positive control worked? Please provide more convincing data.

Line 127) Cite the original publication for Venus (<https://pubmed.ncbi.nlm.nih.gov/11753368/>).

Line 128) Cite the original publication for pUBQ10 (<https://pubmed.ncbi.nlm.nih.gov/8385509/>)

Line 132) Although it is up to the authors how they name their reporter constructs, I find "RGAmPFYR" is a bit difficult to read and pronounce. A simpler name would be desired, e.g. qRGAm2.

Fig 2b) I am astonished that the TagBFP signal of pUBQ10:qRGAmPFYR was not detected in the

root tip, because usually the UBQ10 expression is very strong in that root region, especially in the root cap. Is this due to gene silencing or other effects? Please comment.

Fig. 2) Include full data sets for both pUBQ10:qRGAm2 and pRPS5a:qRGAm2 lines.

Extended Data Fig. 2) Figure panels are not well sorted.

Extended Data Fig 3) Provide quantitative data analyses as in Fig. 2 g,h.

Line 167) Briefly explain why you used 3 - (RGAmPFYR-Venus/TagBFP) as a positive proxy for GA signalling activity.

Fig. 2g and Extended Data Fig. 3 and 4) Add scale bars to all figure panels.

Extended Data Fig. 4 e-i) Which promoter was used in these experiments? In (i) what do the color codes represent?

Line 980) (s, x)?

Line 981) (j) instead of (u)?

Extended Data Fig. 6b) Add scale bar.

Fig 3d and respective main text) Indicate the GA concentration that was used for the treatments in the figure and main text, not only in the caption. Provide representative images for the data in the boxplot, at least in an extended figure.

Figure 3e) Provide representative GA signaling images at least in an extended Figure.

Line 210) Cite the respective reference for the pCLV3::mCherry-NLS reporter if this was not generated in this study.

Fig 4a) I am confused about the labelling in this image, as it indicates the expression of three distinct constructs (pRPS5A::RGAm2-Venus + pRPS5a::TagBFP-NLS + pCLV3::mCherry-NLS) not qRGAm2 together with pCLV3::mCherry-NLS. Furthermore, this figure panel should show the CLV3 expression niche in a single image and the qRGAm2 emission ratio image in a second image. From a single image in which three fluorescence channels were overlayed, it is not clear to me how the authors extracted the GA-signaling map in b.

Extended Data Fig. 8) Revise labeling in a and provide ratio images for qRGAm2.

Fig 4e) I understand that the GPS1 emission ratio scale is set the same as in Extended Data Fig. 9, but it may make sense here to adjust the scale to the min and max ratios to better visualize the distribution of GA in the SAM.

Lines 232-234) For a better understanding, also compare and describe the GPS1 and qRGAm2 patterns with each other.

Figure 5c) Define PCA component 1 and 2.

Figure 5d) Explain better in the figure caption.

Lines 283-286) explain better these numbers. It is not entirely clear to me to which parameters they belong.

Line 1064) "in front and sides of P3". Do you mean IPR around P3. Better indicate this also in the graph, as GA signaling in P3, P4 and P5 is low and the reader might get confused with the P3 etc. labeling. I suggest changing the labels to IPR around P3 etc.

Extended Data Fig. 11) Label in d? Add scale bars to the images in f-h.

Related to Fig. 6) Analyze Venus fluorescence of pCUC2::gai-1-Venus in the SAM to check whether this construct is properly functional in the CUC domain of the SAM.

Lines 383-384) Explain why?

Lines 417-418) Add reference for this statement.

Line 472) To my knowledge, pUBQ10 promoter is only ca. 664 bp, not 2.5 kb.

Line 458-504) Better to provide supplemental tables that list all plasmids and transgenic plant lines that were generated and used in this work.

Lines 516-519) Which pH was used.

Line 521) 50 μ M PAC dissolved in which medium?

Line 566) Add composition of 2x SDS-PAGE buffer.

Lines 640-646) Are these standard Fiji macros? If not, please provide references for the macros.

Supplemental Methods Figure 2) Sorry but I cannot find the description of panel f. Also define and describe better the GA signaling formula ($GA = 3 - RGA/Tag$). Do you mean "GA signaling = 3 - RGAm2-Venus/TagBFP-NLS"? Please explain the "3".

Reviewer #3 (Remarks to the Author):

The manuscript describes the development of a biosensor for gibberellin (GA) action based on a

mutant form of the RGA DELLA protein that is degraded in response to GA signaling. It is mutated to prevent interaction with partner transcription factors such that its presence does not affect GA signaling and plant physiology, but it is degraded by GA, although less effectively than non-mutant RGA. For visualization, the sensor is fused with the VENUS yellow fluorescent protein and co-expressed with nuclear-localised TagBFP, which acts as a reference, the ratio of the two fluorescence outputs allowing GA activity to be quantified in situ. The sensor is used to map GA activity in the Arabidopsis SAM, indicating that highest activity is in the interprimordial regions (IPRs), which corresponded to regions of high expression of the GID1 receptors and accumulation of bioactive GA, measured using the nlsGPS1 sensor. Interestingly, this GA activity in the IPR did not correspond with enhanced cell expansion or cell division, but promoted a transverse plane of division, allowing anisotropic growth that is commensurate with internode elongation from the IPR. Promotion of anisotropic growth is a well established consequence of GA action.

I found this to be a very thorough study that provides one of the most detailed assessments of the role of GA in the SAM. In particular, the sensor is very thoroughly tested to confirm its ability to measure changes in GA signaling and its lack of physiological activity. Impressively, the mutant RGA was tested in Y2H assays with almost all known DELLA interactors. It reinforces the importance of the mutated PFYRE motif in the GRAS domain for the interaction with partner transcription factors. Compared with the tested mutations in other conserved motifs, the selected mutation provided the strongest disruption of interaction with TFs, but conveniently had the least effect on GA-induced degradation. The sensor should prove a useful tool for probing sites of GA action more generally.

The observation that there was no apparent correlation between the strength of GA signaling and enhanced cell expansion or division was unexpected. However, as is pointed out, the morphological observations are focused on the epidermis so do not provide a complete picture. It has been shown for other tissues that there is strong GA signaling in the endodermis, although in relation to elongation growth. However, the described observations are of considerable interest.

I did not find Extended Figure 7 very convincing. Although on the basis of GUS expression and in situ hybridization, GID1b is clearly the most highly expressed GID1 gene in the SAM, from the figure it is unclear that GID1c is expressed, but not GID1a.

On a minor point, chemical names such as paclobutrazol or fluorescein should not be capitalized. In relation to the GA-fluorescein uptake experiment, it has been shown that some transporters do not transport the fluorescein conjugate so its use may not give an accurate picture of GA distribution.

Point-to-point response for all reviewers:

We thank the reviewers for assessing in depth our manuscript, and for their positive and constructive feedback that has helped to significantly strengthen our data. We apologize for the long delay with the previous version. This was due to the fact that we had to generate new plant material to address the concerns that were raised by the referees. Notably, we realized that in our control line with the delta17 mutation in the sensor construct we had lost the fluorescence in the SAM. We first selected new lines and grew them up to the T3 generation only to realize that all the lines were also silenced. To circumvent the problem, we had to generate a new control line (see below) that was only available for analysis (T3 generation) at the beginning of the fall. We describe below all the changes we have made to the text and the many data that we have added to the manuscript. Please note that changes in the text (main text, material and methods and Figure legends) are highlighted in yellow.

Reviewer #1 (Remarks to the Author):

NB. Changes in the text (main text, material and methods and Figure legends) are highlighted in yellow.

Major points:

1. The construction of reporter line is not reliable. First, the choose of one of the DELLA proteins is not sufficiently evidenced. Peptides among DELLA proteins and the structure of whole protein is related to both its two functions, three residues mutation is not convinced. Second, this reporter depends on the degradation of DELLA upon GA sensing, but there are many intermediate steps from GA level to its function. The distribution of receptors, catalytic enzymes and proteasome pathway are all involved. So this reporter line is not a good system to reflect GA signal in vivo.

Reply: We feel that these comments (and a few of the next ones) come from a possible confusion about our goals in the development of the qmRGA sensor (this is the new name of our sensor to address a comment from referee #2). We apologize to the referee if we have not been clear and would like to clarify that we are not claiming that we have developed a reporter that detects GA concentration. On the contrary we aimed at developing a reporter that detects the GA signaling activity resulting from the combined effect of GA and GID1 receptors on the degradation of an inactive DELLA, and that is effectively dependent on other catalytic enzymes and proteasome activity as indicated by the referee. The referee is then absolutely right that our sensor is, by design, 'not a good system to reflect GA signal in vivo' but it is not meant to. Our sensor is complementary to other tools like the nlsGPS1 fret sensor (Rizza et al. Nature Plants 2017) that monitors the GA signal (and we also use this sensor in our work) and allows to understand how cells transform the GA level within a cell into a signaling activity dependent on the many steps rightfully pointed out by the referee (the receptors, proteasome activity etc...) that induce a non-linearity in the treatment of the information (ie the input signaling activity is not a linear function of the GA concentration). Importantly, our sensor actually allows to account for the 'many intermediate steps', whose contribution would be otherwise extremely difficult to evaluate. This was already

indicated in text (Page 6 at the end of the 'Modifying the RGA protein for sensor construction' section) but we further highlighted it more clearly Pages 4 and 5 (text in yellow).

Concerning the first point, we have verified by Y2H, western blot and Luciferase/Renilla transcriptional activity experiments that the mutation of HYY of the RGA protein meets well the two criteria for constructing a degradation-based biosensor: i.e, 1- degradation by GA, 2- no/limited interference with endogenous signaling activity. We further show that expression of qmRGA in planta does not interfere with GA-dependent development and that the sensor responds as expected to both endogenous and exogenous changes in GA levels. We thus believe that we have provided solid and ample evidences that qmRGA is a reliable input signaling sensor (but effectively not a GA concentration sensor, as pointed out by the referee). Considering this and in line with the comment of the reviewer, we have been very careful in the manuscript to always describe our reporter as a GA signaling sensor.

2. In plant, GA level is low and the span is narrow. Plant response to hormone is quite sensitive. The exogenous fluorescent protein GFP/BFP may be not sensitive enough to reflect define GA level.

Reply: Thanks for the comments. We totally agree with the two first sentences. However, concerning the last one, we show that our sensor is able to detect changes in GA signaling activity induced by exogenous GA application (e.g. significant difference 1h after 10 μ M GA application, Fig. 2g-h and Extended Data Fig. 4g-i) and endogenous GA level change (experiments with nitrate, Extended Data Fig. 6), as well as differences within a tissue with a cell definition as for example in the shoot apical meristem. As pointed out in our response to the comment #1 of the reviewer, we would also like to stress that we are not analyzing GA levels with our sensor but the input in the signaling pathway. And the data we provide show that our sensor is sensitive enough to detect changes in signaling input activity caused by both endogenous and exogenous changes in GA concentration.

3. GA concentration and signal should be distinguished.

Reply: We fully agree with the referee and this relates again to point #1. We clearly show (see points #1 and 2) that we are able to monitor changes in GA signaling activity with our sensor. Analyses of GA concentration in root and hypocotyl have already been published using the nlsGPS1 fret sensor (Rizza et al. Nature Plants 2017). For this reason, we used nlsGPS1 (and also fluorescent GAs) to evaluate the contribution of GA concentration distribution to the GA signaling activity distribution (detected using qmRGA) (Fig. 4i). This allows us to distinguish GA concentration and GA signal as proposed by the referee.

4. The concentration of GA (100 μ M) in exogenous treatment is too high.

Reply: In order to evaluate the effects of exogenous GA on the biosensor, sensitivity of the sensor has been tested with 1, 10 and 100 μ M GA (and not only with 100 μ M GA) both in seedling and the SAM (Extended Data Fig. 4a-b, e-f) and 100 μ M gave the

highest response in the SAM. The fact that high concentration of hormones has to be used for the SAM has already been seen for example for auxin (Reinhardt et al. Plant Cell 2000 where auxin concentration in the mM range was used in the SAM).

We then mostly used 100 μ M to analyze the temporality of the response of the sensor to exogenous GA in the SAM in order to have a maximal response (Extended Data Fig. 4h-i). We hope that the referee will agree with us that this was justified and we have added this information in the legend of Extended Data Fig. 4.

5. How about qRGAmPFYR signal in the knock-out mutants GA receptor?

Reply: The triple *gid1* mutant plants are dwarf and sterile, making them difficult to use in such an analysis. To investigate the effect of GA receptors, we have rather expressed a version of our sensor with a delta17 mutation in the RGA sequence. The delta17 mutation is a deletion of a part of the DELLA domain – DELLAVLGYKVRSSSEMA (Dill et al. PNAS 2001). Structural work has shown that this sequence contains a GID1 recognition motif (Murase et al. Nature 2008) and the mutation is sufficient to abolish interaction with GID1 (for GAI: Willige et al. Plant Cell 2007). So, although indirect, this approach is equivalent to analyzing the effect of loss-of-function of the receptors on the degradation of the sensor and we show that introducing the delta17 mutation in the sensor stabilizes it and make insensitive to exogenous GA. We feel that this indeed demonstrates that degradation of qmRGA is dependent upon the GA receptors, answering the request from the referee.

6. WUS and CLV3 should be detected in della mutants to confirm developmental defect in shoot apical meristem.

Reply: Thanks for the suggestion. RNA *in situ* results using *CLV3* in quad and global *della* mutants have been added in a new Extended Data Fig. 13e-g. We also have added whole-mount RNA *in situ* results for *STM* to detect meristematic cells (new Extended Data Fig. 13h-j). Both probes now support the increase in SAM size in *della* mutants as requested by the referee and further show a lateral expansion of the stem cell niche in the mutants.

7. How do GID1a,b,c distribute in L1-3 layers in SAM?

Reply: In Extended data Fig. 7, our results using RNA *in situ* hybridization of GID1a, b, c on SAM longitudinal sections show that there is no GID1a expression in the SAM, and that GID1b and 1c are distributed throughout L1-L2 and L3.

8. The finding that GA does not regulate cell expansion or division but play a role in cell plane orientation is not consistent with previous study. How about GA distribution in cells? Is the change in cell plane orientation caused by unbalance of GA?

Reply: Thanks for the comments and questions. We do not claim that GAs do not regulate cell expansion or division but our correlation and functional analyses rather show that there is a more specific role for GA in controlling the orientation of the plane of cell division in the SAM. So, our results identify a more specific function for GAs in the control on cell division and in the orientation of cell expansion in the SAM, that is

not incompatible with the well-known function for GA in cell division/elongation. For GA cellular distribution, as there are no tools available, we are not able to give an answer. Moreover, we feel that exploring this question is beyond the scope of the manuscript.

9. To explore the contribution of GA signalling to cell division, genetic evidence is missed.

Reply: Thanks for the suggestion. In the revised version, we have added data from *ga2ox* quintuple mutants (*ga2ox1-1*, *ga2ox2-1*, *ga2ox3-1*, *ga2ox4-1* and *ga2ox6-2* combined mutations) as the new Extended Data Fig. 12a,b,c. In this genetic background with higher GA levels, we observed a similar effect on the orientation of cell division than in the *della* mutants (Page 15-16). So, this strengthens our genetic analysis and we now have genetic evidences from 3 different materials (*della* mutants, *pCUC2::gai* transgene plant and *ga2ox quin* mutant) to support the contribution of GA signaling to the control of the orientation of cell division. We feel that this answers the comment of the referee.

10. The influence of GA to cell division is direct or indirect? How about other hormones. Whether GA play a role through other hormones? Is this function of GA similar in other tissues except SAM? Do the GA biosynthetic mutants have similar phenotypes in cell division or it is just accessory role of GA downstream target genes?

Reply: Thanks for the questions. We have mentioned in the discussion that “we cannot eliminate the possibility that this effect could also be indirect, e.g. mediated by GA-induced softening of the cell wall.” (Page 20). In this revised version, we have added to the discussion a phrase on potential interactions of GA with other hormones in the regulation of cell division orientation (highlighted text on page 21). We feel that exploring whether GAs function in cell division orientation is similar in other tissues is out of the scope of this research, but this is definitely an excellent direction to explore in the future. Regarding GA biosynthetic mutants, we refer the referee to point #9: we show that the *ga2ox* quintuple mutant, which accumulates higher levels of GA, has a similar phenotype on cell division orientation in the SAM than the *della* mutants (see Extended Data Fig. 12a,b,c). This strengthens our demonstration that GA regulates cell division orientation.

11. GAI, RGA, RGL GUS lines only show the promoter activity, protein fusion lines should be used instead.

Reply: Thanks for the comment. Providing a comprehensive view of the protein expression of DELLA proteins in the SAM goes beyond our goal in this manuscript. We use the expression data from both GUS lines, *in situ* hybridization and protein fusion lines to show that all DELLAs are expressed in the SAM, which indicates the need to work with multiple DELLA mutants. Also the patterns of the two DELLA proteins (RGL3 and RGA) that we provided in Extended Data Fig. 11i-j already show that the expression of the proteins is compatible with our results with the sensor.

12. In the construction of pCUC2::*gai-1*-VENUS, why chose CUC2? It is not the member with the highest activity in CUC family.

Reply: Thanks for the question and comment. The choice of CUC promoter was not related to its level of activity but rather to its pattern of expression. CUC2 and CUC3 have similar expression pattern in the inflorescence meristem (Hibara et al. Plant Cell 2006) but CUC3 has a greater contribution during embryonic SAM and cotyledon separation (Vroemen et al. Plant Cell 2003). We used the CUC2 promoter to ensure a more specific effect in the inflorescence meristem.

13. Cell division angles are not changes in pCUC2::*gai-1*-VENUS, but frequency is higher, and taken together, the phenotype of SAM in GA mutants implies that GA dose not have direct function on SAM but more likely through other hormones such auxin and cytokinin.

Reply: Thanks for the comment. We actually found that the angles are changed as there is a statistically significant accumulation of 80°-90° divisions (Fig. 6j). The “frequency” in the figure refers to division angle frequency rather than division frequency. So this is not *per se* in contradiction to a direct function of GA in regulating cell division orientation. Of course, we cannot exclude an indirect effect through other hormones and, as pointed out above, we have added to the discussion a phrase on potential interactions of GA with other hormones in the regulation of cell division orientation (highlighted text on page 21).

14. P15, L343-344, “the curvature of the tissue” is phenotype, but not the cause.

Reply: We agree with the referee that the curvature of the SAM is a phenotype and we detected a change in size and curvature that must be induced by the expression of *gai-1* in the CUC2 domain. As argued Page 16, pressure shell models of the SAM (as used in Hamant et al. Science 1998) suggest that such a change in morphology changes the distribution of the mechanical stresses which could thus impact the orientation of cell division (as shown in Louveaux et al. PNAS 2016). However, the referee is right that an alternative hypothesis is that our observations are not explained by the change in morphology due to the change in curvature. Indeed, recent work (Åhl et al. Frontiers Plant Sci 2022) suggests that changes in the regional mechanical properties can then change curvature. The expression of the *pCUC2::gai-1*-VENUS transgene could have such an effect also explaining our observations. We have modified the text Page 17 (highlighted text in yellow) accordingly and now present both hypotheses.

15. Figure7, in the genetic experiment, biosynthetic mutants should be used instead of *della* mutant.

Reply: Thanks for the suggestion. The reviewer is right. As indicated above (points #9&10), besides the *della* mutant, we have thus added new data on cell division orientation in the SAM produced using a biosynthetic mutant (*ga2ox quin*) as the new Extended Data Fig. 12a,b,c. This further strengthens our demonstration.

Minor points:

1. Figure 1, BRE was proved to be a promoter element that bound by BZR1 and repress gene expression downstream BR signaling pathway (Zhiyong Wang, Science, 2005). Here, the results showed activation of target genes, it is conflict with previous reports.

Reply: Thanks for pointing out this potential discrepancy. However, in this experiment we fused the activator VP16 to BZR1 to convert the repressive activity of BZR1 into an activator. So the result is normal and there is no discrepancy. We have added a phrase in the figure legend (highlighted in yellow) to bring attention to this fact.

2. P7, L152, “than” should be changed to “to”.

Reply: Thanks for pointing out this typo. It has been changed as suggested.

3. Typo error of “signaling” all over the manuscript.

Reply: Thanks for the comment. It is not a typo, but the spelling in American English.

Reviewer #2 (Remarks to the Author):

NB. Changes in the text (main text, material and methods and Figure legends) are highlighted in yellow.

In the manuscript by Shi et al. entitled “A quantitative gibberellin signalling biosensor reveals a role for gibberellins in internode specification at the shoot apical meristem”, the authors generated a degradation-based GA signaling reporter using a mutated/modified RGA DELLA protein GA co-receptor and investigated GA signaling in the shoot apical meristem (SAM).

1. First the authors generated four RGA mutant versions and tested them in yeast-2-hybrid assays for disrupted interaction with known interaction partners, but intact interaction with GID1 GA receptors, and for intact GA-induced degradation. Based on their results, RGAm2 was chosen for further characterization. Yeast-2-Hybrid analyses indicated a strong impairment of RGAm2 interaction with RGA interaction partners. Using a transient tobacco infiltration assay it was confirmed that RGAm2 had no effect on BZR1 and TCP transcriptional response in contrast to RGA. Co-IP data of RGA(m2) interaction with IDD2 (Figure 1e) were not convincing.

Reply: Thanks for pointing the need of more convincing data. Following the referee’s advice, we have repeated the Co-IP and the results are presented on Fig. 1e. Furthermore, to strengthen our conclusions, we have also included new Co-IP results obtained with TCP14 on the same Figure panel. We believe that these new results are much more convincing.

2. Based on the above results, the authors generated pUBQ10: or pRPS5a:RGAm2(renamed to RGAmPFYR)-Venus-2A-TagBFP-NLS reporter constructs (in short qRGAmPFYR; I will use here qRGAm2 for simplicity) and generated transgenic Arabidopsis lines. In general, the authors show that qRGAm2 lines are not disturbed in GA responses and respond to GA by VENUS protein level or fluorescence decrease, although not complete datasets for both promoter:qRGAm2 lines are provided (see Fig. 2).

Reply: We actually have checked the decrease of Venus fluorescence upon GA treatment for both promoter lines. The results for pUBQ10 are presented on Fig. 2g (and also Extended Data Fig. 4a,c) but the one for pRPS5a are presented on Extended Fig. 4e,g,h. We presented similar data for the two promoters but in different tissues i.e. the ones for which the promoters provide a homogenous expression through the tissue: hypocotyl for pUBQ10 and SAM for pRPS5a line. We feel that this is sufficient for the characterization of the sensor and we hope that the referee will agree with that.

3. Then the authors analyzed GID1a-c GA-receptor expression patterns in Arabidopsis. The data indicated distinct, but overlapping expression patterns, although the data were limited to analyses in the SAM and in 7-day-old seedlings. More importantly, the authors provided data from hypocotyls, indicating that overexpression of the GA receptor GID1a lead to an enhanced GA signaling reported by qRGAm2.

Furthermore, qd17RGAm2 appeared insensitive to GA in hypocotyls. This line may provide an important negative control, which in theory would be representative for the qRGAm2 GA signaling minimum GA-independent reporter level.

Reply: We agree with the referee that the version of our sensor with the d17 mutation is an important control. To address this comment, we have now expanded the Fig. 3 by adding images showing that the VENUS signal of the d17 line is stable even upon GA addition (Fig. 3f-g). We have attempted unsuccessfully to use this line also as a control in the SAM: see point #4.

4. qRGAm2 (strange image labeling in Fig. 4a) analyses in the SAM indicated that GA signaling decreased during the development of flower primordia (older primordia exhibited lower GA signaling), while GA signaling was high in inter primordial regions (IPR). Complementary analyses using the GA FRET-based Biosensor GPS1 and fluorescent GAs indicated that GAs accumulate in IPR, but not in the central zone (CZ) in which GA signaling (qRGAm2) also appeared relatively high. In these analyses it would have been good if the authors would have conducted parallel analyses with their important negative control reporter qd17RGAm2.

Reply: We fully agree with the referee that using the d17 version of our sensor in the SAM would have been interesting and we attempted to do so. Starting from two lines under the RPS5a promoter that we had used for the manuscript, we analyzed expression in the SAM and could not find any fluorescence. We thus reanalyzed in large number of T1s to obtain lines with expression in the SAM. However, we observed again a complete silencing in T3. Although we could find plants with expression in T2, this expression was variable between SAMs of a given line, possibly due to some level of silencing in different domains of the SAM. We thus could not use any of these lines for our analysis and have no explanation of why this happened.

Given these observations, we decided to generate a new control line where VENUS and TagBFP tagged to NLS are co-expressed under the RPS5a promoter (pRPS5a::VENUS-2A-TagBFP). Here the fluorescence of VENUS is totally independent of GA and, similarly to the d17 line, this new control line allows to control from any bias from the activity of the 2A peptide or other parameters controlling the accumulation of VENUS or TagBFP. We now present new data obtained with this line on Extended Data Fig. 8c-l. This demonstrates a homogenous accumulation of both VENUS and TagBFP throughout the periphery of the meristem. We however could detect a lower VENUS level compared to TagBFP in the central zone. The origin of this effect on the relative production of VENUS and TagBFP is unknown but this observation suggests that the sensor could over-estimate GA signaling activity in this part of the meristem. Given that our focus is primarily in the organogenesis domain of the meristem, this does not modify our conclusions. We have discussed this new control and the corresponding results on Page 10-11 (highlighted in yellow).

Concerning to the comment on strange labelling in Fig. 4a, please see reply to your point #18. We have corrected the labelling to pRPS5a::qmRGA, pCLV3::mCherry-NLS, consistently with the description in the main text.

5. Then the authors compared GA signaling in the SAM with cellular growth in the SAM. These rather complex analyses were at least for me quite difficult to understand given the limited explanation in the main text and the figure caption of Fig. 5. It appeared an anticorrelation of GA signaling and cell expansion, and a low correlation between GA signaling and growth anisotropy. From the data the authors concluded that higher GA signaling in the IPR acts on cell growth orientation rather than expansion. From their data the authors concluded that GA signaling controls the orientation of cell division. This hypothesis was supported by analyses using a global della mutant in which GA signaling is strongly increased. Conversely, suppression of GA signaling in the IPR via *pCUC2::gai-1-Venus* expression rather affected division plane angles in non-IPR cells and the curvature of the SAM. In this respect the authors should clearly check via fluorescence analyses in the SAM if the *pCUC2::gai1-Venus* construct is properly expressed in the SAMs CUC2 domain.

Reply: Thanks for the comments and suggestions. We have checked Venus expression in the line we used and the data have been added in the new Extended Data Fig. 14b. We thus show now that the *pCUC2::gai1-Venus* construct is properly expressed, as requested by the referee. We have indicated this in the text Page 16 (text highlighted in yellow).

6. Finally, the authors showed some SAM data to indicate the influence of GA signaling (via genetic manipulation) on internode organization. As I am not an expert of SAM, I was not able to follow these explanations and presented data. Also, I suggest that the authors conduct similar timelapse SAM analyses (as in Fig. 7) using their *qRGAm2* and *qd17RGAm2* reporters to support their claims, especially given the title of the manuscript. Adding the global della mutant to Extended Data Fig. 14 would also be helpful.

Reply: We feel that this comment of the reviewer may be due to the fact that we had not clearly described our goals in this part of the manuscript. We have amended our text on Page 8 to clarify the fact that we follow the cellular organization of cells in the IPR in the vicinity of a primordia from stage 1/2 to stage 4. This corresponds to the time during which GA signaling increases in the IPR as indicated by the analysis of the *qmRGA* pattern (Fig. 4). We have completed our data by providing on Extended Data Fig. 8c-f and k new data showing the *qmRGA* pattern over time (t=0 and 24h). This shows that the GA signaling pattern detected by *qmRGA* pattern remains consistent over a time, suggesting that we effectively analyze cells with a higher GA signaling activity over the time of our analysis (that goes to 34h i.e. a bit longer). Thus, this approach allows us to analyze the effect of the increase in GA signaling activity on the cellular organization of the IPR in wild-type and mutants, and to show that higher GA signaling contributes to organize the IPR in parallel cell files, as seen in internodes. We hope that these changes make the explanations easier to follow.

For the reasons provided in point #4 we have not been able to use the sensor line with the *d17* mutation. We have rather used the new *pRPS5a::VENUS-2A-TagBFP* as a control in this experiment. We provide new data on Extended Data Fig. 8g-i, l that shows that while high GA signaling detected by the sensor is associated to the

internode precursor cells, the VENUS/TagBFP fluorescence ratio is rather homogenous over time in the entire organogenesis domain in the control line. This strengthens our demonstration that the changes reported by the sensor are associated to specification of internode precursor cells.

As requested, the global *della* mutant images have been added to Extended Data Fig. 15a-b.

Other comments:

1. Lines 98-99) Define the exact mutations that were introduced in RGA^{m1} to RGA^{m4}, as this is not shown in Figure 1 and Extended Data Figure 1) or define them in the respective figures.

Reply: As requested, the exact mutations introduced in RGA^{m1} to RGA^{m4} are now described in the main text (Page 5), in Fig. 1 and Extended Data Fig. 1 and in the respective figure legends.

2. Figure 1a and extended Data Figure 1a) Indicated mutated amino acid numbers.

Reply: We have corrected as requested.

3. Figure 1e) In the co-IP experiment RGA-GFP with IDD2-RFP, the positive control is barely detectable. How can the authors conclude from such data that the RGA^{m2}-GFP interaction with IDD2-RFP is abolished when not even the positive control worked? Please provide more convincing data.

Reply: We agree with the reviewer, the interaction between RGA-GFP and IDD2-RFP was weak in the previous version of our manuscript. As mentioned above, to strengthen this result, we have repeated the Co-IP and have also included an additional control, TCP14. In the new Fig. 1e we show that the interactions of RGA^{m2} with IDD2 and TCP14 are substantially reduced compared to RGA.

4. Line 127) Cite the original publication for Venus.

Line 128) Cite the original publication for pUBQ10 (<https://pubmed.ncbi.nlm.nih.gov/8385509/>)

Reply: Thanks for the suggestions. We have corrected and cited the original references in the new version.

5. Line 132) Although it is up to the authors how they name their reporter constructs, I find "RGA^{mPFYR}" is a bit difficult to read and pronounce. A simpler name would be desired, e.g. qRGA^{m2}.

Reply: To follow the suggestion of the referee, we have chosen a simpler name and now call our sensor qmRGA throughout the manuscript.

6. Fig 2b) I am astonished that the TagBFP signal of pUBQ10:qRGA^{mPFYR} was not detected in the root tip, because usually the UBQ10 expression is very strong in that root region, especially in the root cap. Is this due to gene silencing or other effects? Please comment.

Reply: The distance from ATG of UBQ10 to its upstream gene is 2.5kb and we have used all this entire intergenic region as was indicated in the material and method. In the literature, different sizes have been used: 634bp (Norris et al. 1993); 1307 bp (Wang et al. 2021); 1584bp (Michniewicz et al. 2015). We suspect that there are motifs located between -1.5kb to -2.5 kb upstream of UBQ10 which introduces a repression of pUBQ10 expression in the root tip, explaining our observation. We have now indicated Page 6 that it is a 2.5 kb promoter to attract attention to this fact.

7. Fig. 2) Include full data sets for both pUBQ10:qRGAm2 and pRPS5a:qRGAm2 lines.

Reply: We have actually shown full data sets of changes in Venus fluorescence intensity upon GA treatment for both promoter lines (pUBQ10: Fig. 2g, Extended Data Fig. 4a,c; pRPS5a: Extend Data Fig. 4e,g,h). Please see response to your point #2.

8. Extended Data Fig. 2) Figure panels are not well sorted.

Reply: Thanks for the advice. We have rearranged the disposition of the panels, as suggested.

9. Extended Data Fig 3) Provide quantitative data analyses as in Fig. 2 g,h.

Reply: Thanks for the comment. We now show quantitative analyses in Extended Data Fig.3b,d,f.

10. Line 167) Briefly explain why you used $3 - (\text{RGAmPFYR-Venus}/\text{TagBFP})$ as a positive proxy for GA signalling activity.

Reply: To quantify the GA signalling activity from qmRGA, we compute a ratio between VENUS and TagBFP fluorescence, where TagBFP accounts for promoter activity. Since mRGA-VENUS is degraded upon GA treatment, mRGA-Venus fluorescence is negatively related to GA signaling, while $3 - \text{VENUS}/\text{TagBFP}$ is positively related to GA signaling and filters possible variations due to promoter activity changes. We then used " $3 - (\text{mRGA-Venus}/\text{TagBFP})$ " as a positive proxy for GA signaling that fully covers the range of values in VENUS/TagBFP fluorescence ratio that we measure and that is quite variable notably in hypocotyls. Fig. 2h, Extended Data Fig. 4b and Extended Data Fig. 6c illustrate this range of values: the data are from hypocotyls and values on the y axis ranges from 0 to 3.

We now explain this better on Page 8 (highlighted text). We also refer the readers to Supplementary method 1, paragraph 1.2.4 where this is also explained.

11. Fig. 2g and Extended Data Fig. 3 and 4) Add scale bars to all figure panels.

Reply: Thanks for the suggestions. Scale bars have been added to all figure panels in Fig. 2g and Extended Data Fig. 3 and 4.

12. Extended Data Fig. 4 e-i) Which promoter was used in these experiments? In (i) what do the color codes represent?

Reply: It was the RPS5a promoter used in these experiments. In (i), the red and green boxes represent GA and mock treatment, respectively and this has now been indicated

in the figure. We have also added and highlighted all these information in the figure legend.

13. Line 980) (s, x)?

Line 981) (j) instead of (u)?

Reply: Thanks for spotting this. (s, x) has changed to (f, g). (u) has changed to (j).

14. Extended Data Fig. 6b) Add scale bar.

Reply: This has been done.

15. Fig 3d and respective main text) Indicate the GA concentration that was used for the treatments in the figure and main text, not only in the caption. Provide representative images for the data in the boxplot, at least in an extended figure.

Reply: Thanks for the suggestion. The GA concentration is now indicated in the figure and main text, and the representative images are shown in the new Fig. 3d.

16. Figure 3e) Provide representative GA signaling images at least in an extended Figure.

Reply: As requested, the representative images are now shown in the new Fig. 3f.

17. Line 210) Cite the respective reference for the pCLV3::mCherry-NLS reporter if this was not generated in this study.

Reply: Thanks for the suggestion. The reference is Ma et al., elife 2019 and has been added Page 10 in the revised version.

18. Fig 4a) I am confused about the labelling in this image, as it indicates the expression of three distinct constructs (pRPS5A::RGAm2-Venus + pRPS5a::TagBFP-NLS + pCLV3::mCherry-NLS) not qRGAm2 together with pCLV3::mCherry-NLS. Furthermore, this figure panel should show the CLV3 expression niche in a single image and the qRGAm2 emission ratio image in a second image. From a single image in which three fluorescence channels were overlaid, it is not clear to me how the authors extracted the GA-signaling map in b.

Reply: Thanks for the comments and suggestions. In the revision, we now show four images besides the pRPS5a::qmRGA, pCLV3::mCherry-NLS overlay image in the new Fig. 4b-e. These are pRPS5a::qmRGA, the mRGA-VENUS channel, the TagBFP channel and pCLV3::mCherry-NLS, respectively. We have also corrected the labelling of these five images to avoid confusion.

19. Extended Data Fig. 8) Revise labeling in a and provide ratio images for qRGAm2.

Reply: Thank for the suggestions. We have changed the label to pRPS5a::qmRGA and have provided ratio images for qmRGA.

20. Fig 4e) I understand that the GPS1 emission ratio scale is set the same as in Extended Data Fig. 9, but it may make sense here to adjust the scale to the min and max ratios to better visualize the distribution of GA in the SAM.

Reply: Thanks for the suggestion. We have now changed the min and max ratio scale from (0.8, 1.6) to (0.8, 1.4) of the GPS1 image in order to have a stronger color contrast.

21. Lines 232-234) For a better understanding, also compare and describe the GPS1 and qRGAm2 patterns with each other.

Reply: Thanks for the suggestion. We have added Page 11 one sentence to compare and describe the GPS1 and qRGAm2 patterns with each other.

22. Figure 5c) Define PCA component 1 and 2.

Reply: PCA component 1 is mostly associated to negatively weighted “Surfacic growth intensity” and positively weighted “GA signaling” and PCA component 2 to positively weighted “Surfacic growth anisotropy” and negatively weighted “CLV3 expression” (with minor contributions from the other two variables in both cases). The percentage near each component represents the proportion in which the variations of the data along the component’s direction contribute to the total variability of the data. We have added this information to the legend of Figure 5.

23. Figure 5d) Explain better in the figure caption.

Reply: We revised the figure caption as below: ‘Spearman correlation study between GA signaling, surfacic growth intensity and surfacic growth anisotropy at tissue level but excluding the CZ. The numbers on the right are Spearman's rho values between the two associated variables. Stars indicate that the correlation/anti-correlation is highly significant.’

24. Lines 283-286) explain better these numbers. It is not entirely clear to me to which parameters they belong.

Reply: Thanks for the comments. We have explained better Page 13 about the angle and percentage numbers. To help the readers, we have also added a new Fig. 5h illustrating the circumferential/transverse orientation.

25. Line 1064) “in front and sides of P3”. Do you mean IPR around P3. Better indicate this also in the graph, as GA signaling in P3, P4 and P5 is low and the reader might get confused with the P3 etc. labeling. I suggest_changing the labels to IPR around P3 etc.

Reply: Thank you for the suggestion. We have changed the labels to IPR (cells) around P3/P4/P5 as suggested. The main text and figure legends were also revised accordingly.

26. Extended Data Fig. 11) Label in d? Add scale bars to the images in f-h.

Reply: *rgl2* is a gene trap line with Ds-GUS insertion in the exon (Lee et al. 2002). This line can be used as a GUS reporter line to show the expression of RGL2. We have

revised the label in d to “*rgl2* (*Ds-GUS* insertion line)”. Scale bars have been added to images in f-h.

27. Related to Fig. 6) Analyze Venus fluorescence of pCUC2::*gai1*-Venus in the SAM to check whether this construct is properly functional in the CUC domain of the SAM.

Reply: Thanks for the suggestion. VENUS expression was analyzed and shows that the pCUC2::*gai1*-VENUS construct is properly expressed in the CUC domain of the SAM. Images with Venus and cell wall propidium iodide staining have been added in the new Extended Data Fig. 14b.

28. Lines 383-384) Explain why?

Reply: We now explain better in the text Page 18 that the fact that IPR cells are located between successive primordia and divide following a pattern that organize cells in radial files as in internodes suggest that these cells are internode precursor cells. We have revised the sentence by adding an explanation.

29. Lines 417-418) Add reference for this statement.

Reply: Thanks for this suggestion. A reference has been added.

30. Line 472) To my knowledge, pUBQ10 promoter is only ca. 664 bp, not 2.5 kb.

Reply: Please see our response to your point #6. In literature, different lengths of UBQ10 promoter have been used. According to Norris et al. 1993, the length of UBQ10 promoter is 634bp. But Wang et al. 2021 used 1307 bp upstream of ATG as the promoter, and Michniewicz et al. 2015 used 1584bp and detected whole-seedling expression including root tip. The distance from ATG of UBQ10 to its nearest upstream gene is 2.5kb. That's why we used this promoter length. To avoid any confusion about this, we have clearly indicated in the text that we are using a 2.5 kb promoter (see point #6).

31. Line 458-504) Better to provide supplemental tables that list all plasmids and transgenic plant lines that were generated and used in this work.

Reply: We have followed the suggestion of the reviewer and provided a new supplemental table 2 with this information.

32. Lines 516-519) Which pH was used.

Reply: Thanks for noticing this omission. The pH used was 5.8. We have added this information in the text (highlighted on page 24).

33. Line 521) 50 μ M PAC dissolved in which medium?

Reply: Thanks for noticing this omission. Pac was dissolved in ethanol to make a 50 mM stock and diluted in water to make a 50 μ M working solution. We have added this information in the text (highlighted on page 24).

34. Line 566) Add composition of 2x SDS-PAGE buffer.

Reply: Thanks for the suggestion. To simplify, we have replaced 2x SDS-PAGE buffer by protein extraction buffer, and the composition of the buffer is indicated on Page 26.

35. Lines 640-646) Are these standard Fiji macros? If not, please provide references for the macros.

Reply: Almost all of these macros use the plugin "3D ImageJ suite" (Ollion et al. 2013) of Fiji. We have revised Page 29 the sentence for macros and added the reference for the plugin "3D ImageJ suite".

36. Supplemental Methods Figure 2) Sorry but I cannot find the description of panel f. Also define and describe better the GA signaling formula ($GA = 3 - RGA/Tag$). Do you mean "GA signaling = 3 – RGA_{m2}-Venus/TagBFP-NLS"? Please explain the "3".

Reply: Indeed, the panel (f) was not described in the figure legend, we corrected this. Regarding the formula, please see our response to your point #10. In the legend, we refer to paragraph 1.2.4 of Supplemental Methods 1 where this is now explained more clearly (in addition to the main text).

Reviewer #3 (Remarks to the Author):

NB. Changes in the text (main text, material and methods and Figure legends) are highlighted in yellow.

The manuscript describes the development of a biosensor for gibberellin (GA) action based on a mutant form of the RGA DELLA protein that is degraded in response to GA signaling. It is mutated to prevent interaction with partner transcription factors such that its presence does not affect GA signaling and plant physiology, but it is degraded by GA, although less effectively than non-mutant RGA. For visualization, the sensor is fused with the VENUS yellow fluorescent protein and co-expressed with nuclear-localised TagBFP, which acts as a reference, the ratio of the two fluorescence outputs allowing GA activity to be quantified in situ. The sensor is used to map GA activity in the Arabidopsis SAM, indicating that highest activity is in the interprimordial regions (IPRs), which corresponded to regions of high expression of the GID1 receptors and accumulation of bioactive GA, measured using the nlsGPS1 sensor. Interestingly, this GA activity in the IPR did not correspond with enhanced cell expansion or cell division, but promoted a transverse plane of division, allowing anisotropic growth that is commensurate with internode elongation from the IPR. Promotion of anisotropic growth is a well established consequence of GA action.

I found this to be a very thorough study that provides one of the most detailed assessments of the role of GA in the SAM. In particular, the sensor is very thoroughly tested to confirm its ability to measure changes in GA signaling and its lack of physiological activity. Impressively, the mutant RGA was tested in Y2H assays with almost all known DELLA interactors. It reinforces the importance of the mutated PFYRE motif in the GRAS domain for the interaction with partner transcription factors. Compared with the tested mutations in other conserved motifs, the selected mutation provided the strongest disruption of interaction with TFs, but conveniently had the least effect on GA-induced degradation. The sensor should prove a useful tool for probing sites of GA action more generally.

The observation that there was no apparent correlation between the strength of GA signaling and enhanced cell expansion or division was unexpected. However, as is pointed out, the morphological observations are focused on the epidermis so do not provide a complete picture. It has been shown for other tissues that there is strong GA signaling in the endodermis, although in relation to elongation growth. However, the described observations are of considerable interest.

Reply: Thanks for the very positive comments on our work.

1. I did not find Extended Figure 7 very convincing. Although on the basis of GUS expression and in situ hybridization, GID1b is clearly the most highly expressed GID1 gene in the SAM, from the figure it is unclear that GID1c is expressed, but not GID1a.

Reply: We agree with the referee that images of shoots after GUS staining can be difficult to analyze. To focus and highlight the difference at the SAM, we have now added arrows in the GUS images (Extended Data Fig. 7a-c) where *GID1a* has no expression (the SAM is white) but *GID1c* has expression (the SAM is stained in light blue). The *in situ* hybridization results in the figure are coherent with this conclusion and we hope that the referee will agree that they provide two complementary pieces of data that support our conclusions.

2. On a minor point, chemical names such as paclobutrazol or fluorescein should not be capitalized.

Reply: Thanks for this comment. We have corrected this.

3. In relation to the GA-fluorescein uptake experiment, it has been shown that some transporters do not transport the fluorescein conjugate so its use may not give an accurate picture of GA distribution.

Reply: We agree with the referee. The GA-fluorescein has definitely limitations but this is why we also used the FRET biosensor nlsGPS1. The fact that GA-fluorescein gives a consistent pattern with nlsGPS1 (higher GA levels at the boundaries from P4) suggest that GA-fluorescein is transported at least in part similarly to endogenous GA. We thus feel that using GA-fluorescein in conjunction with nlsGPS1 is providing useful information and an array of evidence supporting a differential distribution of GA within the SAM. In the text (highlighted Page 11), we added 'As a complementary approach,' before describing the results to further stress the fact that GA-fluorescein and nlsGPS1 are used in conjunction.

Reviewer #4 (Remarks to the Author):

In this study, the authors attempted to develop a reporter for detecting GA signaling using RGA, one of the DELLA proteins. In RGAm2, which has mutations in three amino acid residues of the PFYRE domain, it was demonstrated that while it shows GA-dependent degradation, the original GA response inhibitory function of RGA is suppressed. The expression of qmRGA utilized two promoters, and these results highlight the necessity of choosing promoters based on the tissue being observed. The effects of qmRGA on the plant body have been shown to be very limited. The authors have clarified GA levels using GA-FI and nlsGPS1, and consistent with this, it was revealed that GA signaling is high in IPR. Furthermore, it has been suggested that GA signaling in IPR is involved in controlling the orientation of the cell division plane.

The qmRGA presented in this study is more stable than native RGA, yet it exhibits quick responsiveness to exogenous GA. Through techniques such as Y2H, transient assays, and investigations into plant phenotypes, the central functions of qmRGA are felt to be adequately explained. While this research concentrates on GA signaling and the cell division plane in the epidermis surrounding the SAM, it is anticipated that future applications in other tissues will enhance the overall understanding of GA signaling throughout the plant.

Below I have a number of suggestions that the authors may consider in the further process of improving the manuscript.

Unlike the visualization techniques for gibberellin (GA) concentration that have been reported in recent years, such as GA-FI and nlsGPS sensors, this study's focus on the development of a visualization marker for GA signaling is recognized for its novelty. However, visualization of DELLA proteins using RGA-GFP has been perceived as already conducted. Therefore, the novelty of visualizing DELLA proteins with this study's qmRGA may feel less apparent to readers. It would be beneficial to clearly demonstrate the novelty of this study compared to previous research using RGA-GFP in the introduction.

In this study, the authors report through Y2H that RGAm2, with three amino acid substitutions in the PFYRE domain, cannot bind with many interacting factors that have been reported previously. This verification has been conducted thoroughly and is commendable. However, it has been reported that factors like BZR1 and PIF bind to the LHRI domain, and IDD2/GAF1 binds to the SAW domain. While this study demonstrates that the three amino acids in the PFYRE domain are very important for the interaction between RGA and its interacting factors, questions remain regarding why these three amino acid substitutions affected the binding to RGA of interacting factors that have been reported to bind to other domains. Although the importance of the PFYRE domain is mentioned in the discussion highlighted in yellow, clarifying the reason for selecting these three amino acids for mutation in RGAm2 within the text could resolve such questions.

Reviewer #5 (Remarks to the Author):

The authors succeeded in the very challenging task to design a GA sensor that provides a meaningful readout on a signaling response, while not or barely interfering with the signaling itself. The authors have clearly described the possibilities that are offered by the novel sensor, but also carefully examined and explained its limitations

Using their novel sensor, the authors detect low GA signaling activity in the SAM center and elevated GA activity in the IPR in floral primordia. Very interestingly, evidence is provided that differential GA activities in the SAM and IPR do not have an effect on cell division rates but rather act on cell division plane positioning. The conclusions are supported by several pharmacological and genetic approaches, including GA overaccumulation and in perception- and signaling-defective mutants. In their discussion, now alternative mechanisms of action are discussed, which are clearly tasks for future studies.

I reviewed the manuscript for the first time and focused on assessing whether concerns by previous reviewers have been addressed satisfactorily. All in all, the authors have undertaken great effort to answer questions, mitigate concerns and address criticisms, which resulted in a very

thoroughly controlled and convincing study, with conclusions, which are sufficiently carefully phrased and of broad interest. Thereby the authors show that their novel GA-activity sensor has the potential to substantially advance our understanding of the complex hormonal signaling networks that drive plant development.

Addressing concerns about discrepancies between known hormonal signal amplitudes and the output range of the sensor, the authors clearly described that the sensor is not a GA-level indicator but a signaling sensor. The dynamic range of such a sensor design therefore reflects the range of a signaling response, which certainly can – and most likely typically is – larger than the initial signal amplitude. To link GA distribution and GA signaling activity, the authors present independent experiments, including an GA-fluorescein uptake assay and measurements involving a previously published nlsGPS1 biosensor.

The use of 100 μM GA in exogenous applications remains certainly a concern as such high concentrations are beyond physiological. Nevertheless, these using high concentrations for treatments can be justified as it appears to be necessary to reach meaningful interference intracellularly (due to penetration efficiency?). It may be worth discussing this issue in the article more prominently, as other readers might wonder as well.

The authors provide several new datasets to support their findings, such as data on CLV3 expression through in situ hybridization, *ga2ox* data, new data on cell division orientation, and data on the *d17* mutation for the hypocotyl. They provided improved data for their CoIP experiments and addressed concerns on GID1 distribution, responded to an apparent controversy on the role of GA in cell expansion, addressed concerns regarding pCUC2::*gai1*-Venus expression, and convincingly explained their choice of promoter activity versus protein fusion to measure expression, as well as their choice of the CUC2 promoter. Importantly, the authors provided improved explanations of the goals behind some of the experiments, since the story has overall become quite complex.

In the rebuttal, the authors are transparent about their attempts to analyze the *d17* mutant in the SAM, which was not possible due to technical reasons that appear to be beyond their control. It might be considered to add this information to the manuscript to help others learn from their failed attempt. As a suitable and elegant alternative, the authors introduce a new pRPS5a::*VENUS*-2A-TagBFP reporter line, which controls for parameters such as 2A peptide cleavage and relative fluorophore intensity.

Line 189: I suggest to replace the phrase “all other tissues tested” with something more concrete like “epidermis of hypocotyl and outer cell layers of root tips”.

Spelling: in response to one reviewer it was explained that American English was used, hence, the spelling of “signaling”. However, in other instances British spelling was used; e.g. “centre”. Please check and correct.

Point-to-point response to reviewers:

We thank both reviewers for taking the time for this final evaluation of our manuscript and for their very positive feedback. Our responses to their remaining concerns can be found below.

Reviewer #4 (Remarks to the Author):

In this study, the authors attempted to develop a reporter for detecting GA signaling using RGA, one of the DELLA proteins. In RGA_{m2}, which has mutations in three amino acid residues of the PFYRE domain, it was demonstrated that while it shows GA-dependent degradation, the original GA response inhibitory function of RGA is suppressed. The expression of qmRGA utilized two promoters, and these results highlight the necessity of choosing promoters based on the tissue being observed. The effects of qmRGA on the plant body have been shown to be very limited. The authors have clarified GA levels using GA-FI and nlsGPS1, and consistent with this, it was revealed that GA signaling is high in IPR. Furthermore, it has been suggested that GA signaling in IPR is involved in controlling the orientation of the cell division plane.

The qmRGA presented in this study is more stable than native RGA, yet it exhibits quick responsiveness to exogenous GA. Through techniques such as Y2H, transient assays, and investigations into plant phenotypes, the central functions of qmRGA are felt to be adequately explained. While this research concentrates on GA signaling and the cell division plane in the epidermis surrounding the SAM, it is anticipated that future applications in other tissues will enhance the overall understanding of GA signaling throughout the plant.

Below I have a number of suggestions that the authors may consider in the further process of improving the manuscript.

Unlike the visualization techniques for gibberellin (GA) concentration that have been reported in recent years, such as GA-FI and nlsGPS sensors, this study's focus on the development of a visualization marker for GA signaling is recognized for its novelty. However, visualization of DELLA proteins using RGA-GFP has been perceived as already conducted. Therefore, the novelty of visualizing DELLA proteins with this study's qmRGA may feel less apparent to readers. It would be beneficial to clearly demonstrate the novelty of this study compared to previous research using RGA-GFP in the introduction.

Reply: To address this remaining concern, we have now modified the text in the introduction on Page 4 and Page 5 to further highlight the novelty of our approach. We now point out on Page 4 that, while it has been extremely valuable, the use of RGA-GFP fusion expressed under their own promoter for mapping GA signaling activity is biased by the fact that the RGA promoter is differentially expressed through the plant and within tissue. The differential activity of the RGA promoter is expected to contribute to the fluorescence pattern, thus making the approach not quantitative. On page 5, we now also point out that our biosensor uses a mutated version of RGA that can be expressed ubiquitously in order to monitor GA signaling activity across tissues without interfering with the GA signaling activity. This would not have been possible with RGA-GFP.

In this study, the authors report through Y2H that RGA_{m2}, with three amino acid substitutions in the PFYRE domain, cannot bind with many interacting factors that have been reported previously. This verification has been conducted thoroughly and is commendable. However, it has been reported that factors like BZR1 and PIF bind to the LHRI domain, and IDD2/GAF1 binds to the SAW domain. While this study demonstrates that the three amino acids in the PFYRE domain are very important for the interaction between RGA and its

interacting factors, questions remain regarding why these three amino acid substitutions affected the binding to RGA of interacting factors that have been reported to bind to other domains. Although the importance of the PFYRE domain is mentioned in the discussion highlighted in yellow, clarifying the reason for selecting these three amino acids for mutation in RGAm2 within the text could resolve such questions.

Reply: We had already given information about how the amino-acids to mutate were chosen in the results, 1st paragraph (Page 6). We believe that we clearly highlight that the RGAm2 mutation was selected based on the work by Hirano et al. 2010 (Ref 23 in the manuscript) which shows that a HYY497AAA mutation in the rice DELLA protein SLR1 (which is equivalent to RGAm2 mutation) inhibits its growth repressing activity, while it only slightly reduces its GA-induced degradation ability. It is still unclear why this mutation interferes with the ability of the transcription factor partners to interact with the DELLAs (that usually binds through the LHRI and/or SAW motif); the PFYRE motif could stabilize the interaction. We have now added this information in the Discussion to complement what was already in the manuscript (Page 20,21).

Reviewer #5 (Remarks to the Author):

The authors succeeded in the very challenging task to design a GA sensor that provides a meaningful readout on a signaling response, while not or barely interfering with the signaling itself. The authors have clearly described the possibilities that are offered by the novel sensor, but also carefully examined and explained its limitations

Using their novel sensor, the authors detect low GA signaling activity in the SAM center and elevated GA activity in the IPR in floral primordia. Very interestingly, evidence is provided that differential GA activities in the SAM and IPR do not have an effect on cell division rates but rather act on cell division plane positioning. The conclusions are supported by several pharmacological and genetic approaches, including GA overaccumulation and in perception- and signaling-defective mutants. In their discussion, now alternative mechanisms of action are discussed, which are clearly tasks for future studies.

I reviewed the manuscript for the first time and focused on assessing whether concerns by previous reviewers have been addressed satisfactorily. All in all, the authors have undertaken great effort to answer questions, mitigate concerns and address criticisms, which resulted in a very thoroughly controlled and convincing study, with conclusions, which are sufficiently carefully phrased and of broad interest. Thereby the authors show that their novel GA-activity sensor has the potential to substantially advance our understanding of the complex hormonal signaling networks that drive plant development.

Addressing concerns about discrepancies between known hormonal signal amplitudes and the output range of the sensor, the authors clearly described that the sensor is not a GA-level indicator but a signaling sensor. The dynamic range of such a sensor design therefore reflects the range of a signaling response, which certainly can – and most likely typically is – larger than the initial signal amplitude. To link GA distribution and GA signaling activity, the authors present independent experiments, including an GA-fluorescein uptake assay and measurements involving a previously published nlsGPS1 biosensor.

The use of 100 μ M GA in exogenous applications remains certainly a concern as such high concentrations are beyond physiological. Nevertheless, these using high concentrations for

treatments can be justified as it appears to be necessary to reach meaningful interference intracellularly (due to penetration efficiency?). It may be worth discussing this issue in the article more prominently, as other readers might wonder as well.

Reply: Application of 100 μ M GA may seem high, but is routinely used in the literature when GA is sprayed on adult plants or during short-term kinetics, in order to visualize a fast and significant response. We would like to emphasize that we also applied lower concentrations of GA (1 μ M and 10 μ M GA), and the responsiveness is still visible (Supplementary Figure 4). As suggested, we have added a phrase in the main text to discuss this issue when describing the results of Figure 3 (Page 10).

The authors provide several new datasets to support their findings, such as data on CLV3 expression through in situ hybridization, ga2ox data, new data on cell division orientation, and data on the d17 mutation for the hypocotyl. They provided improved data for their CoIP experiments and addressed concerns on GID1 distribution, responded to an apparent controversy on the role of GA in cell expansion, addressed concerns regarding pCUC2::gai1-Venus expression, and convincingly explained their choice of promoter activity versus protein fusion to measure expression, as well as their choice of the CUC2 promoter. Importantly, the authors provided improved explanations of the goals behind some of the experiments, since the story has overall become quite complex.

In the rebuttal, the authors are transparent about their attempts to analyze the d17 mutant in the SAM, which was not possible due to technical reasons that appear to be beyond their control. It might be considered to add this information to the manuscript to help others learn from their failed attempt. As a suitable and elegant alternative, the authors introduce a new pRPS5a::VENUS-2A-TagBFP reporter line, which controls for parameters such as 2A peptide cleavage and relative fluorophore intensity.

Reply: We had already indicated in the manuscript that we observed silencing of qmRGA but we have now added extra information on Page 11 in the following way:
“While the *qd17mRGA* construct was systematically silenced in the SAM of T3 plants of 5 independent lines we characterized in depth,”

Line 189: I suggest to replace the phrase “all other tissues tested” with something more concrete like “epidermis of hypocotyl and outer cell layers of root tips”.

Reply: This has been corrected as suggested on Page 9.

Spelling: in response to one reviewer it was explained that American English was used, hence, the spelling of “signaling”. However, in other instances British spelling was used; e.g. “centre”. Please check and correct.

Reply: We apologize for overseeing this. This has been checked and corrected.